# Klf5 regulates muscle differentiation by directly targeting muscle-specific genes in cooperation with MyoD in mice

Shinichiro Hayashi[1], Ichiro Manabe[2], Yumi Suzuki[1], Frédéric Relaix[3], Yumiko Oishi[1]*

[1]Department of Cellular and Molecular Medicine, Medical Research Institute, Tokyo Medical and Dental University, Tokyo, Japan; [2]Department of Aging Research, Graduate School of Medicine, Chiba University, Chiba, Japan; [3]INSERM U955 IMRB-E10 UPEC, ENVA, EFS, Creteil, France

**Abstract** Krüppel-like factor 5 (Klf5) is a zinc-finger transcription factor that controls various biological processes, including cell proliferation and differentiation. We show that Klf5 is also an essential mediator of skeletal muscle regeneration and myogenic differentiation. During muscle regeneration after injury (cardiotoxin injection), Klf5 was induced in the nuclei of differentiating myoblasts and newly formed myofibers expressing myogenin *in vivo*. Satellite cell-specific *Klf5* deletion severely impaired muscle regeneration, and myotube formation was suppressed in *Klf5*-deleted cultured C2C12 myoblasts and satellite cells. *Klf5* knockdown suppressed induction of muscle differentiation-related genes, including myogenin. Klf5 ChIP-seq revealed that Klf5 binding overlaps that of MyoD and Mef2, and Klf5 physically associates with both MyoD and Mef2. In addition, MyoD recruitment was greatly reduced in the absence of Klf5. These results indicate that Klf5 is an essential regulator of skeletal muscle differentiation, acting in concert with myogenic transcription factors such as MyoD and Mef2.

*For correspondence: yuooishi-circ@umin.ac.jp

**Competing interests:** The authors declare that no competing interests exist.

## Introduction

Skeletal muscle, the dominant organ for locomotion and energy metabolism, has a remarkable capacity for repair and regeneration in response to injury, disease and aging. Regeneration of adult skeletal muscle following injury occurs through the mobilization of satellite cells (SCs), a population of injury-sensitive quiescent muscle stem cells that activate, proliferate, differentiate, and fuse with injured myofibers. (*Buckingham and Rigby, 2014*; *Relaix and Zammit, 2012*). In an adult skeletal muscle, SCs, characterized by expression of the paired box transcription factor Pax7 are mitotically quiescent and reside in a niche between the basal lamina and sarcolemma of associated muscle fibers. In response to muscle damage, SCs are activated and assume a myoblast identity, thereby initiating muscle repair. After massive proliferation, SC-derived myoblasts undergo differentiation and fusion to form myotubes that replace the damaged myofibers (*Comai and Tajbakhsh, 2014*), while a subset of activated satellite cells downregulate MyoD, exit the cell cycle to replenish the muscle stem cell pool.

A large body of studies have shown that a family of four myogenic regulatory factors (MRFs) regulating early skeletal muscle development also regulate the postnatal muscle regeneration program. These MRFs include the myogenic basic-helix-loop-helix type transcription factors MyoD and Myf5, which are recruited to skeletal muscle-specific genes regulatory regions, where they determine myogenic fate and initiate the differentiation cascade. Thereafter, MyoD increases the expression of late target genes in cooperation with myogenin and MRF4 through a feed-forward regulatory mechanism

that regulates terminal differentiation (*Penn et al., 2004*). MRFs also enhance myogenic activity by interacting with other transcription factors, including Mef2 (*Fong and Tapscott, 2013*; *Molkentin and Olson, 1996*).

Although the core MRF network that governs skeletal muscle regeneration and differentiation has been identified, the sequence of events within muscle-specific gene regulatory regions during differentiation is not fully understood. It is highly likely, for example, that the myogenic program driven by MRFs requires input from other transcription factors. More specifically, MRFs, including MyoD, may associate with other transcription factors that respond to microenvironmental cues and restrict the binding of MRFs to particular subsets of target enhancers. This would enable MRF complexes to context dependently regulate muscle genes governing the processes of myocyte differentiation and regeneration. Identification of these MRF partners is therefore important to clarify the mechanism by which MRF-containing transcriptional regulatory complexes achieve precise spatiotemporal regulation of their target genes.

Krüppel-like factors (Klfs) are a subfamily of zinc-finger transcription factors. All Klf proteins contain a DNA-binding domain consisting of three zinc fingers positioned at their carboxyl-terminal end, which enables their specific binding to 'GT-box' or 'CACCC' sites (*Bieker, 2001*). Several Klf members are involved in the control of skeletal muscle differentiation and function (*Prosdocimo et al., 2015*). For instance, Klf3 is upregulated during skeletal muscle differentiation and transactivates the muscle creatin kinase (*Mck*) promoter (*Himeda et al., 2010*). Klf2 and Klf4 are also upregulated in differentiating muscle cells and promote muscle cell fusion (*Sunadome et al., 2011*). Klf6 promotes and Klf10 inhibits myoblast proliferation (*Dionyssiou et al., 2013*; *Parakati and DiMario, 2013*). Klf15 critically regulates skeletal muscle nutrient catabolism, contributing to the regulation of exercise capacity and muscle wasting (*Gray et al., 2007*; *Haldar et al., 2012*; *Shimizu et al., 2011*). These studies demonstrate the pivotal involvement of Klfs in muscle biology. However, their role in the global transcriptional program of muscle differentiation remains unclear. Interestingly, Cao et al. previously showed that binding sites for Sp1, which is structurally related to Klfs and binds to similar GC-rich sequences, are highly enriched in MyoD-binding regions, which suggests there may be interaction between these two transcription factors.

Klf5 is reportedly involved in embryonic development (*Shindo et al., 2002*), control of cellular proliferation and differentiation (*Fujiu et al., 2005*; *Oishi et al., 2005*), stress response (*Shindo et al., 2002*), and apoptosis (*Zhu et al., 2006*). *Klf5* null embryos fail to develop beyond the blastocyst stage *in vivo* or to produce embryonic stem cell lines *in vitro*, indicating that Klf5 is essential for maintenance of stemness (*Ema et al., 2008*; *Shindo et al., 2002*). Klf5 is expressed in many cell types, including gut epithelial cells (*Chanchevalap et al., 2004*; *Sun et al., 2001*), vascular smooth muscle cells (*Fujiu et al., 2005*), adipocytes (*Oishi et al., 2005*), neuronal cells (*Yanagi et al., 2008*) and leukocytes (*Yang et al., 2003*). In smooth muscle cells, Klf5 regulates the genes involved in early processes of differentiation, including SMemb/NMHC-B and SM22α (*Adam et al., 2000*; *Watanabe et al., 1999*). Klf5 is also required for adipocyte differentiation. In 3T3-L1 preadipocytes, Klf5 is induced at an early stage of differentiation by C/EBPβ and δ, which is followed by expression of PPARγ$_2$ (*Oishi et al., 2005*). We previously reported that Klf5 acts as a regulator of lipid metabolism in skeletal muscle (*Oishi et al., 2008*). However, the role of Klf5 in skeletal muscle differentiation has not been characterized. In the present study, therefore, we investigated the role of Klf5 in skeletal muscle differentiation and regeneration.

## Results

### Klf5 is highly expressed in differentiating myoblasts during muscle regeneration

To test the hypothesis that Klf5 is involved in skeletal muscle regeneration, we first examined tibialis anterior (TA) muscle from C57BL/6J wild-type mice with and without cardiotoxin (CTX)-induced skeletal muscle injury (*Lepper et al., 2009*). As compared to expression in intact (uninjured) muscle, expression of *Klf5* mRNA in regenerating TA muscles was upregulated 2 and 4 days after injury, as was expression of myogenic factors (*Pax7, Myod1, Myog* and *Myh3*) (*Figure 1A*).

Our immunohistochemistry results showed that little or no Klf5 is expressed in Pax7-positive, quiescent SCs or myonuclei that constitute intact muscle (*Figure 1B* upper panel, *Figure 1—*

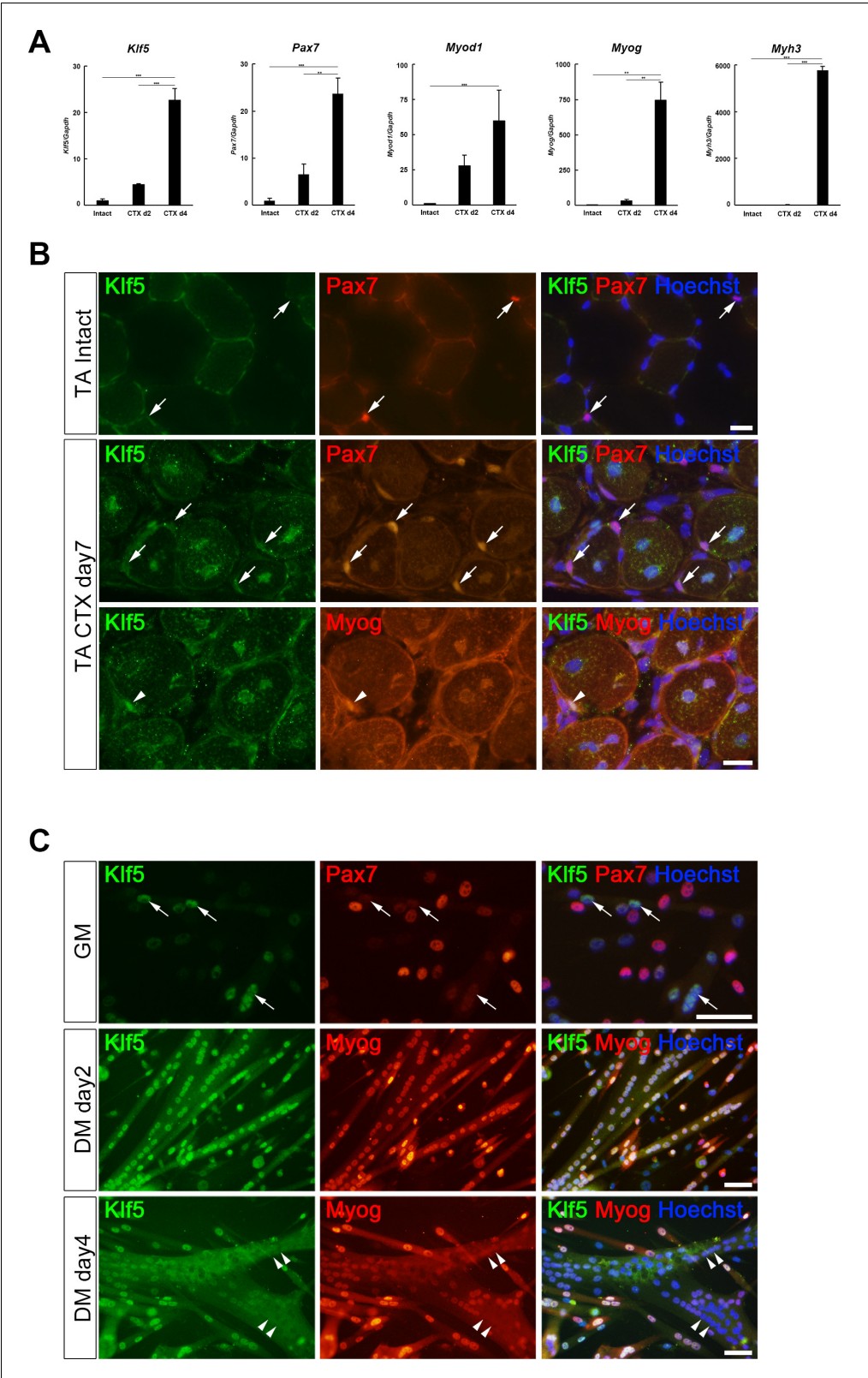

**Figure 1.** Klf5 is upregulated during myogenesis. (**A**) Relative mRNA expression of *Klf5, Pax7, Myod1, Myog* and *Myh3* (embryonic myosin heavy chain) in intact and regenerating TA muscles. The animals were sacrificed on day 4 after CTX injection. Data are means ± SEM.
***p<0.001. **p<0.01. Representative data from three individual mice are shown. (**B**) Klf5 expression during muscle regeneration. Klf5 was not detectable in quiescent satellite cells (SCs) in intact muscle (arrows in the top panels). During muscle regeneration, Klf5 was highly expressed in the
*Figure 1 continued on next page*

*Figure 1 continued*

nuclei of differentiating myocytes expressing Myog (arrowheads) and regenerating myofibers, whereas Klf5 expression was not detected or detected very weakly in Pax7-positive SCs (arrows). Representative data from at least three individual mice are shown. Scale bar represents 20 µm. (**C**) Plated SCs were cultured in growth medium (GM) or differentiating medium (DM; for 2 days or 4 days) after isolation and were co-immunostained for Klf5 with Pax7 or Myog. Klf5 is expressed in the differentiating myocytes, which were negative or very weakly positive for Pax7 (arrows). Klf5 was upregulated during differentiation and frequently co-localized with Myog. After 4 days of culture, Klf5 levels were decreased in the nuclei of large myotubes (arrowheads). Representative data from at least three individual mice are shown. Scale bar represents 50 µm.

The following figure supplement is available for figure 1:

**Figure supplement 1.** Preparation of SC primary cultures.

*figure supplement 1A*). Seven days after CTX-induced muscle injury, we observed ongoing muscle regeneration characterized by the presence of centrally located nuclei in regenerating fibers. At that time, expression of Klf5 remained very low in Pax7-positive, activated SCs (*Figure 1B*, middle panel, arrows). However, higher levels of Klf5 were detected in differentiating myocytes, which expressed myogenin 7 days after injury (*Figure 1B*, lower panel, arrowheads). Klf5 was also detected within the centrally located nuclei of regenerating myofibers. These results indicate that Klf5 is markedly upregulated in differentiating myocytes and regenerating myofibers during the course of muscle regeneration *in vivo*.

To examine Klf5 expression during muscle differentiation *in vitro*, SCs were isolated from the extensor digitorum longus (EDL) muscle from wild-type mice and placed in culture. After the first passage, nearly 100% of cells were MyoD positive, confirming the high purity of the muscle cell cultures (*Figure 1—figure supplement 1B,C*). Over 70% of the purified cells were also positive for Pax7 (*Figure 1—figure supplement 1C*). Consistent with the *in vivo* results, little or no Klf5 was detected in a fraction of the activated SCs expressing Pax7 and MyoD (*Figure 1C*, upper panel). Approximately 75% of Pax7-positive cells were also weakly positive for Klf5 (*Figure 1—figure supplement 1D*). Klf5 was induced in differentiating myoblasts exhibiting an absence or low Pax7 expression (*Figure 1C*, arrows, *Figure 1—figure supplement 1D*). After differentiation for 2 days, Klf5 was strongly expressed in the nuclei of differentiating myotubes and co-localized with myogenin (*Figure 1C*, middle panel, *Figure 1—figure supplement 1E*). However, the strong Klf5 expression was transient, and by day 4 of differentiation it was undetectable in the nuclei of larger, fused myotubes that had also lost myogenin expression (*Figure 1C*, arrowheads). These *in vitro* observations indicate that Klf5 is induced in differentiating myoblasts during muscle differentiation and then downregulated in mature myotubes.

## Klf5 is required for muscle regeneration

The finding of temporal overlap between Klf5 expression and muscle regeneration and differentiation *in vivo* and *in vitro* prompted us to examine the role of Klf5 in SC-mediated muscle regeneration *in vivo*. Mice carrying floxed *Klf5* alleles with loxP sites were crossbred with mice expressing *Pax7*-driven CreERT2, a tamoxifen-inducible recombinase to generate $Pax7^{CE/+};Klf5^{flox/flox}$ mice. The $Pax7^{CE/+}$ mice were previously shown to be able to ablate almost 100% of SCs when crossed with $R26R^{GFP-DTA/+}$ mice (*Lepper et al., 2009, 2011*). The $Pax7^{CE/+};Klf5^{flox/flox}$ model enabled us to selectively delete *Klf5* from SCs after tamoxifen injection. To induce SC-specific *Klf5* deletion, 8- to 12-week-old $Pax7^{CE/+};Klf5^{flox/flox}$ mice were treated with tamoxifen for 5 days before use in subsequent experiments. Control $Pax7^{+/+};Klf5^{flox/flox}$ mice were also treated with tamoxifen. Hereafter, $Pax7^{CE/+};Klf5^{flox/flox}$ and $Pax7^{+/+};Klf5^{flox/flox}$ mice will represent the mice treated with tamoxifen in this manner. Muscle histology revealed that $Pax7^{CE/+};Klf5^{flox/flox}$ mice or control $Pax7^{+/+};Klf5^{flox/flox}$ mice displayed no obvious phenotypic alterations under physiological conditions and exhibited no morphological alterations in their skeletal muscle stained with hematoxylin and eosin (H&E) 28 days after tamoxifen treatment (*Figure 2—figure supplement 1*).

After continuous tamoxifen administration for 5 days, CTX was injected into the TA muscles of both $Pax7^{CE/+};Klf5^{flox/flox}$ and control $Pax7^{+/+};Klf5^{flox/flox}$ mice, which were then analyzed 4, 7 and 28 days after injury. On day 4 after injury, mRNA expression of *Myog* and *Myh3* was significantly lower in regenerating muscles from $Pax7^{CE/+};Klf5^{flox/flox}$ mice than control mice; *Pax7* and *Myod1* levels

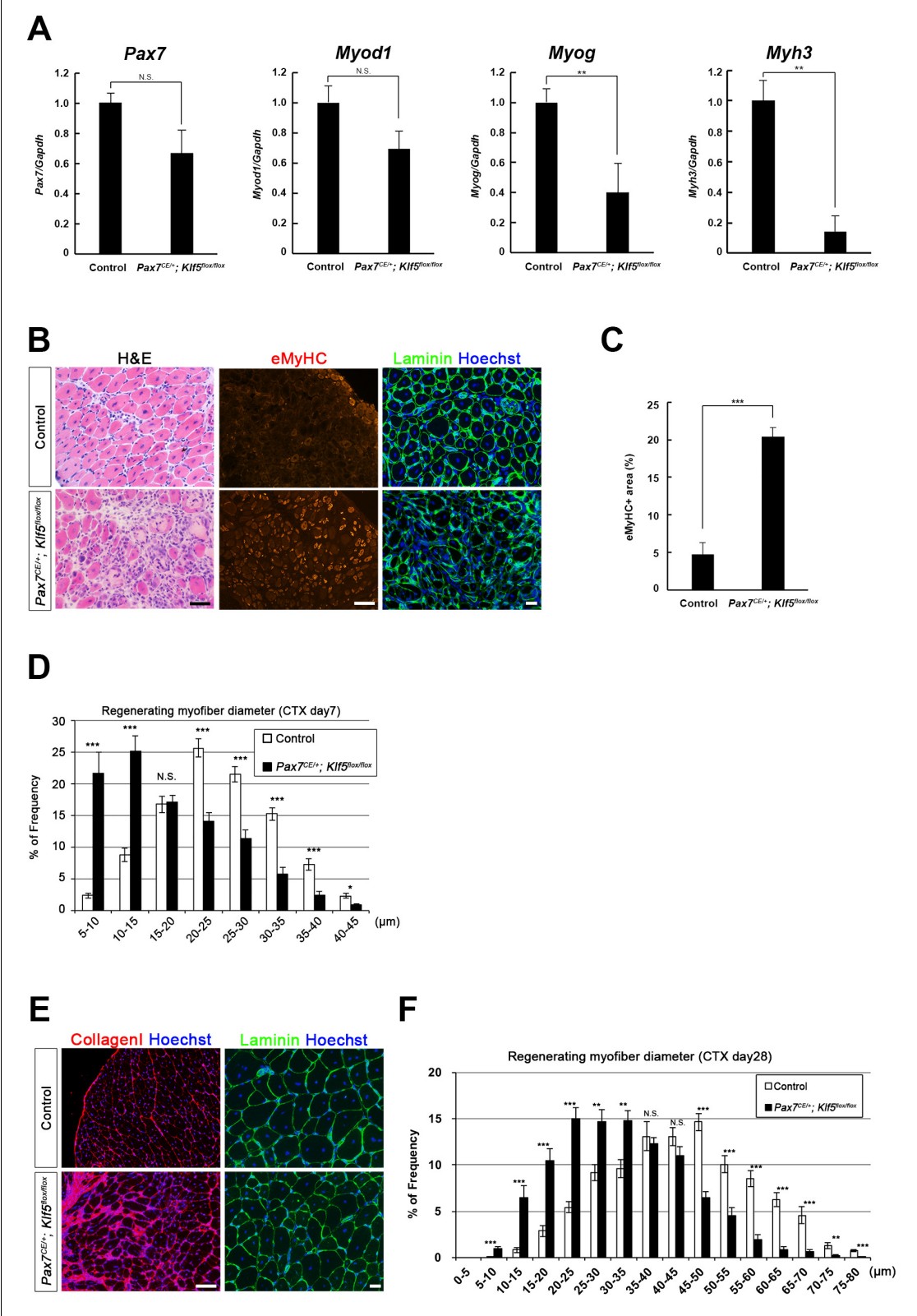

**Figure 2.** Klf5 is required for muscle regeneration *in vivo*. CTX was injected into the TA muscles of SC-specific *Klf5* knockout mice (*Pax7*<sup>*CE/+*</sup>;*Klf5*<sup>*flox/flox*</sup>) and control *Pax7*<sup>*+/+*</sup>;*Klf5*<sup>*flox/flox*</sup> mice. The animals were then sacrificed on day 4 (**A**), 7 (**B–D**) or 28 (**E–F**) after CTX injection. (**A**) *Pax7, Myod1, Myog* and *Myh3* expression in regenerating TA muscle (on day 4 after CTX injection) was analyzed using qRT-PCR (**A**). Data represent means ± SEM. n = 3 for each group. **p<0.01. N.S., not significant. (**B–F**) Sections were stained with H&E or immunostained for eMyHC, laminin and collagen I. Representative

*Figure 2 continued on next page*

Figure 2 continued

images of muscle sections on day 7 (B) or 28 (E) after CTX injection are shown. The eMyHC-positive area 7 days after CTX injection was quantified using Olympus cellSense Digital Imaging software (C). Distributions of myofiber diameters on days 7 (D) and 28 (F) after CTX injection are shown. Representative data from four *Klf5* knockout and five control (for day 7) or four mice for each genotype (for day 28) are shown. Scale bars represent 100 μm. ***p<0.001. **p<0.01. N.S., not significant.
The following figure supplement is available for figure 2:

**Figure supplement 1.** Tamoxifen treatment does not affect TA muscle morphology.

also tended to be reduced, but the differences did not reach statistical significance (*Figure 2A*). On day 7 after injury, H&E and immunofluorescent staining in control mice revealed efficient muscle regeneration characterized by regenerated myofibers with decreased expression of eMyHC, a marker of immature myofibers (*Figure 2B*). On the contrary, the regeneration process was significantly impaired in *Pax7$^{CE/+}$;Klf5$^{flox/flox}$* mice. These mice exhibited necrotic fibers and infiltrating inflammatory cells, and an extensive area of regenerating myofibers with prolonged expression of eMyHC (*Figure 2B and C*). Analysis of the distribution of myofiber diameters revealed that *Klf5* deletion in SCs significantly shifted regenerating myofibers towards smaller diameters (*Figure 2D*), suggesting differentiation and/or maturation of muscle fibers were impaired by the loss of Klf5.

Twenty-eight days after CTX injury in control mice, muscle regeneration was nearly complete, as evidenced by the presence of regenerated, mature fibers and disappearance of inflammatory cells, as previous described (*Ingalls et al., 1998*). However, the regeneration process was retarded in the injured muscle of *Pax7$^{CE/+}$;Klf5$^{flox/flox}$* mice, and massive deposition of collagen I, indicating enhanced fibrosis, was observed (*Figure 2E*). In addition, the regenerating myofibers were still shifted toward smaller diameters (*Figure 2F*). These results indicate that Klf5 is required for proper SC-mediated muscle regeneration after skeletal muscle injury.

## Klf5 is required for myoblast differentiation

The molecular mechanism responsible for muscle regeneration mimics that for skeletal muscle differentiation, and the compromised muscle regeneration after SC-selective *Klf5* deletion suggested to us that Klf5 is necessary for proper skeletal muscle cell differentiation. To test that idea, we used C2C12 myoblasts, a widely-employed cell line that retains the potential to differentiate into myotubes. *Klf5* mRNA was induced after induction of C2C12 cells differentiation and reached a peak on day 4 of myoblast differentiation (*Figure 3—figure supplement 1A*). Klf5 protein was detected after 2 days of differentiation (*Figure 3—figure supplement 1B*) and was frequently colocalized with myogenin in the nuclei of differentiating myotubes, which is consistent with the pattern observed during myogenic differentiation of SCs (*Figure 3—figure supplement 1B* and *Figure 1C*).

To test the requirement for Klf5 in myoblast differentiation, we generated *Klf5*-null C2C12 cells using a CRISPR-Cas9 system (*Ran et al., 2013*). Three *Klf5*-null clones and three control clones were established (*Figure 3—figure supplement 2A*), and it was confirmed that both alleles were targeted by deletion and/or frame-shift mutation at the target loci in the *Klf5*-null clones (*Figure 3—figure supplement 2B*). Klf5 protein was not detectable after induction of differentiation in any of the three *Klf5*-null clones (*Figure 3A* and *Figure 3—figure supplement 2A*), and myotube formation was significantly impaired as compared to the control clones (*Figure 3* and *Figure 3—figure supplement 2A*). This effect occurred concomitantly with decreased expression of myogenin and MyHC on day 4 of differentiation (*Figure 3A–C*). To determine whether forced expression of Klf5 could rescue the compromised differentiation of *Klf5*-null clones, Klf5 was exogenously introduced using a retrovirus harboring murine Klf5 (RV-Klf5). *Klf5*-null C2C12 cells transfected with RV-Klf5 successfully formed myotubes when stimulated to differentiate, whereas *Klf5*-null cells infected with an empty retrovirus failed to do so (*Figure 3D*). In addition, immunohistochemistry and western blotting demonstrated that expression of MyHC and myogenin was partially rescued by the forced expression of Klf5 (*Figure 3D–F*). Klf5 thus appears to be required for a myoblast differentiation.

This requirement for Klf5 in myoblast differentiation was further confirmed by knocking down *Klf5* using specific small interfering RNA (siRNA) in C2C12 cells. Immunohistochemistry indicated that the *Klf5* knockdown efficiently inhibited the induction of myogenin (*Figure 3—figure supplement 3A*

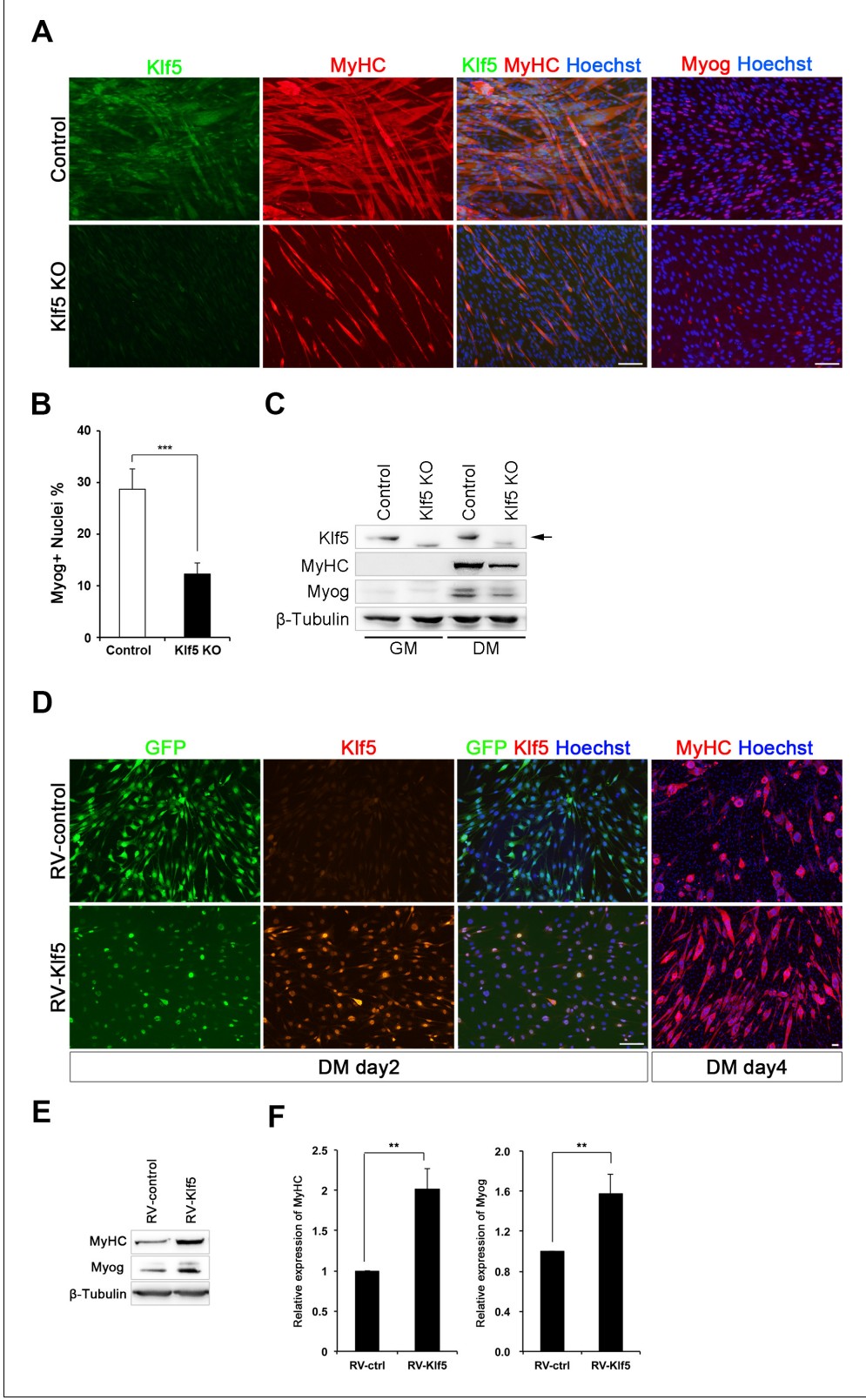

**Figure 3.** Klf5 is essential for muscle differentiation. (**A**) Establishment of *Klf5* knockout (KO) myoblasts using a CRISPR-Cas9 system. *Klf5* KO C2C12 cells or Control cells were immunostained for MyHC and Klf5 or myogenin. *Klf5* KO cells do not express Klf5 during muscle differentiation and exhibit severely reduced myotube formation. MyHC: myosin heavy chain, Myog: myogenin. Scale bar represents 100 μm. (**B**) Percentage of Myog-positive cells among total cells. *Klf5* KO cells exhibited less Myog expression than Control. Data are means ± SEM. (***p<0.001) Representative data from at least

*Figure 3 continued on next page*

*Figure 3 continued*

three individual experiments are shown. (**C**) Western blots showing reduced MyHC and Myog expression in *Klf5* KO C2C12 cells during differentiation. Representative data from at least three individual experiments are shown. (**D**) *Klf5* KO cells were infected with a Retro-viral vector (RV-Klf5) harboring *Klf5* or empty vector (RV-control), after which differentiation was induced for 2 or 4 days. The cells were then fixed and immunostained for Klf5 or MyHC. Impairment of myotube formation, as evidence from the loss of MyHC expression in *Klf5* KO cells was rescued by exogenous Klf5 expression. Representative data from at least three individual experiments are shown. (**E–F**) Western blots revealing the reduction of MyHC and Myog expression in *Klf5* KO C2C12 cells and its rescue by RV-Klf5. The expression levels were normalized to β-Tubulin (**F**). Data represent means ± SEM. (**p<0.01) Representative data from at least three individual experiments are shown.

The following figure supplements are available for figure 3:

**Figure supplement 1.** *Klf5* knockdown impairs myoblast differentiation.
**Figure supplement 2.** Establishment of *Klf5* knockout C2C12 cells.
**Figure supplement 3.** Phenotype of *Klf5* knockdown cells.

*and B*). At the same time, myotube formation was significantly impaired in the *Klf5* knockdown cells, as quantified based on the Fusion index, which represents the fraction of MyHC-positive, fused myotubes that contain at least three nuclei (*Figure 3—figure supplement 3C and D*). Although previous reports have shown the involvement of Klf5 in controlling cell proliferation in several cancer cell lines (*Sun et al., 2001*), *Klf5* knockdown did not affect C2C12 cell proliferation (*Figure 3—figure supplement 3E*).

Because previous studies have shown cooperation and competition among Klf members (*Sunadome et al., 2011*), we analyzed the effects of *Klf5* knockdown on expression of other Klf members. Levels of *Klf6, Klf13, Klf15*, and *Klf16* expression were modestly, but significantly, increased in *Klf5* knockdown cells on day 3 (*Figure 3—figure supplement 1C*). On the other hand, expression of *Klf10* and *Klf11* was decreased in *Klf5* knockdown cells.

## Klf5 is required for induction of myogenesis-related genes

The observation that Klf5 is required for myoblast differentiation in both SCs and C2C12 cells led us to examine the consequences of *Klf5* knockdown on a genome-wide scale. A series of RNA-seq analyses of differentiating myoblasts and myotubes transfected with siRNA targeting Klf5 or control RNA were performed. In line with the suppressed myoblast differentiation, expression of a set of genes involved in myogenesis was significantly decreased in *Klf5* knockdown cells (*Figure 4A*). Among the 12,744 expressed transcripts (normalized counts >4), 1472 with RefSeq annotations were reduced by >1.5-fold on day 3 in the *Klf5* knockdown cells as compared to the control cells. Gene ontology analysis revealed that the genes whose expression was reduced by *Klf5* knockdown had significant enrichment of functional annotations for myogenesis and muscle cell development (*Figure 4B*). Among the 934 genes exhibiting a >1.5-fold increase in expression in *Klf5* knockdown cells, no gene sets directly related to myogenesis were significantly enriched as compared to control cells on day 3 (*Figure 4—source data 1*).

We next assessed the effects of *Klf5* knockdown on MyoD-regulated genes. Analysis of MyoD binding sites using previously reported MyoD ChIP sequencing (ChIP-seq) of C2C12 myotubes differentiated for 2 days (*Cao et al., 2010*) revealed that among the Refseq genes, 753 that contained nearby sites bound by MyoD were upregulated by >two-fold on day 3 as compared with day 0. These genes are most likely direct MyoD targets and are highly enriched with myogenesis-related genes. The effect of *Klf5* knockdown on the expression of MyoD-regulated genes became more pronounced as the course of differentiation proceeded (*Figure 4C*), and expression of those MyoD-regulated genes on day 3 was significantly decreased by *Klf5* knockdown (*Figure 4C and D*). This distinct pattern of gene expression is exemplified by *Myog* and *Myl4* (*Figure 4E*). Quantitative PCR analysis confirmed that expression of MyoD-regulated genes, as exemplified by *Myog, Mybph, Myl4* and *Myom2*, was significantly decreased by *Klf5* knockdown in C2C12 cells on day 3 post-differentiation (*Figure 4F*). This indicates that Klf5 is required for expression of these MyoD-regulated genes. Remarkably, *de novo* motif analysis revealed that the top motifs in the MyoD-bound cistrome

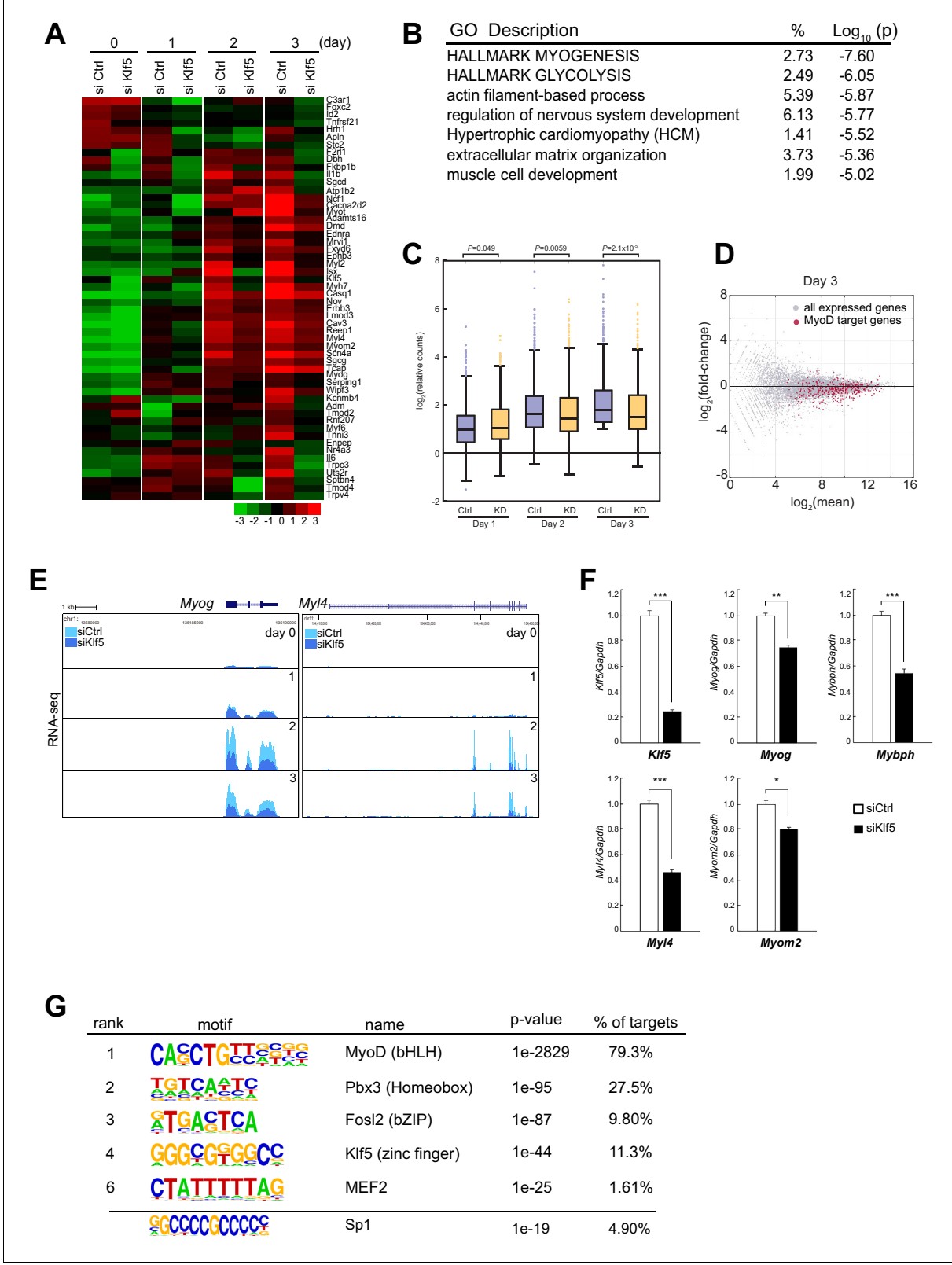

**Figure 4.** Transient *Klf5* knockdown results in a repression of the myogenic program. (**A**) Hierarchical clustering and heatmap of the expression levels (log2 normalized counts) of genes involved in the Hallmark Myogenesis (hallmark gene sets M5909) in C2C12 myoblasts transfected with control or *Klf5*-specific siRNA after induction of differentiation for the indicated times. (**B**) Functional annotations associated with genes that were inhibited in *Klf5* knockdown cells. (**C**) Relative distribution of RNA-seq tags of genes with MyoD binding during the course of myoblast differentiation. Box-and-whisker

*Figure 4 continued on next page*

*Figure 4 continued*

plot showing log2 counts of the MyoD-regulated genes and the remaining RefSeq genes normalized to the counts in control siRNA-transfected cells on day 0. P values are between control siRNA (Ctrl) and siKlf5 (KD) cells analyzed using the Mann-Whitney *U* test. (**D**) MA plot depicting the relationship between fold changes in myotube differentiation-related genes, comparing RNA-seq from differentiating myoblasts transfected with control siRNA or siRNA targeting Klf5. Red dots represent genes with MyoD binding, and the Gray dots represents genes without MyoD binding. (**E**) UCSC genome browser images illustrating normalized RNA-seq reads for the representative myogenesis-related genes *Myog* and *Myl4* in control or Klf5 siRNA-transfected C2C12 cells at the indicated days after induction of differentiation. (**F**) Relative mRNA expression of *Klf5, Myog, Mybph, Myl4* and *Myom2* in C2C12 cells transfected with control siRNA or siRNA targeting Klf5. Data represent means ± SEM. (*p<0.05, **p<0.01, ***p<0.001, n = 3, biological replicates). (**G**) *de novo* motifs identified in regions bound by MyoD in myotubes. The enrichment for a known Sp1 motif was also analyzed.

The following source data is available for figure 4:

**Source data 1.** Gene ontology analysis on the genes upregulated by *Klf5* knockdown.

---

included ones closely matched with known consensus myogenin (E-box), Mef2 and Klf5 binding motifs, which suggests potential co-localization of MyoD, Mef2 and Klf5 at MyoD-regulated enhancers (*Figure 4G*). Cao et al. previously showed that the motif for Sp1 is enriched in regions bound by MyoD (*Cao et al, 2010*). The previously identified Sp1 motif contains a core 5'-CCGCCC-3' sequence, which matches a subset of the Klf motif ranked as 4th in our de novo motif analysis. This suggests the Sp1 motif may also be enriched. Indeed, enrichment analysis of a known Sp1 motif showed that the Sp1 motif is enriched in the MyoD binding regions (*Figure 4G*).

## Klf5 regulates myogenesis-related genes in concert with MyoD and Mef2

The observation that the MyoD-bound cistrome was significantly enriched in the KLF5 motif led us to test the hypothesis Klf5 regulates genes involved in myogenesis in concert with MyoD and Mef2. To investigate the location of Klf5 during the course of myoblast differentiation, we performed ChIP-seq using a Klf5 specific antibody. In addition, we used published ChIP-seq data to analyzed the binding sites of Mef2D in C2C12 myotube on day 5, the major Mef2 isoform involved in expression of late muscle-specific genes (*Sebastian et al., 2013*). As expected, Klf5 bound to the regulatory regions of myogenesis-related genes, as exemplified by *Myog, Myod1, Myl4* and *Mybph*, through-out the course of differentiation (*Figure 5A* and *Figure 5—figure supplement 1*). MyoD and Mef2D binding was frequently observed at those gene loci. The recruitment of Klf5, MyoD and Mef2 to the *Myog* enhancer was further confirmed using ChIP-QPCR (*Figure 5—figure supplement 2A–C*).

*Figure 5B* shows a heatmap of Klf5, MyoD, and Mef2D binding around the top 1000 Klf5 binding peaks in C2C12 myotubes on day 5. Approximately half of the Klf5 peaks had significant MyoD bind-ing peaks in close proximity. An association with Mef2D was also observed in many Klf5 binding regions. Consistent with these findings, genome-wide binding profiles showed that there is signifi-cant overlap among the Klf5, MyoD and Mef2D cistromes (*Figure 5C and D*). Among 4382 Klf5 peaks observed on day 5 post-differentiation, 757 loci (17.2%) were also bound by both MyoD and Mef2D, and another 1467 sites (33.4%) were bound by either MyoD or Mef2D (*Figure 5E*). More-over, Klf5 binding to the MyoD and Mef2D binding peaks increased as differentiation progressed (*Figure 5C and D*, compare day 0 vs. day 2 and 5). Co-immunoprecipitation showed that Klf5 directly associated with MyoD and Mef2 *in vitro* (*Figure 5F*). These results suggest that Klf5, MyoD and Mef2 work together to drive expression of a set of genes involved in myogenic differentiation.

Only a small fraction of the regions that had Klf5 binding before differentiation (day 0) had signifi-cant MyoD binding in close proximity after starting differentiation into myotubes (*Figure 5—figure supplement 3*). This is in contrast to the Klf5-bound regions on day 5 (*Figure 5B*), which showed strong association with MyoD. The association of Klf5 with loci at which Klf5 was already bound before starting differentiation (day 0) was decreased or unchanged over the course of differentiation (*Figure 5—figure supplement 3*). This was in sharp contrast to the association of Klf5 recruited after starting myogenic differentiation (*Figure 5B*). Collectively then, a majority of regions that increas-ingly acquire Klf5 binding during differentiation are also bound by MyoD in proximity to the Klf5 within myotubes. On the other hand, at many observed Klf5 binding regions in myoblasts, binding is decreased during differentiation and not associated with MyoD binding within myotubes. These

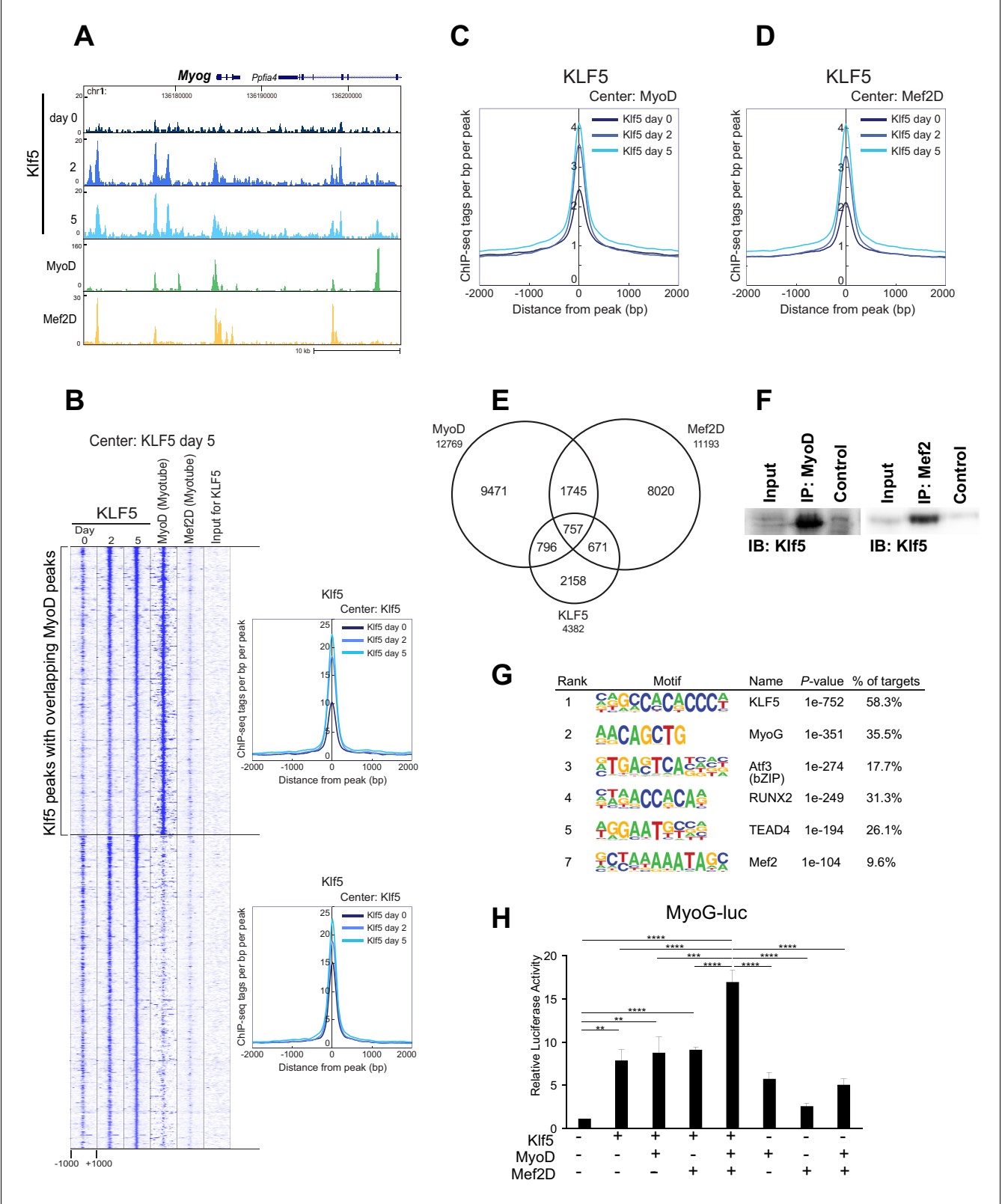

**Figure 5.** Klf5 regulates myogenesis-related target genes in concert with MyoD. (**A**) UCSC genome browser images illustrating normalized tag counts for Klf5 (post-differentiation day 0, 2, 5), MyoD (myotube, differentiated for 2 days) and Mef2D (myotube, differentiated for 5 days) at *Myog* loci in differentiating C2C12 myotubes. (**B**) Heatmap for binding of Klf5 (post-differentiation day 0, 2, and 5), MyoD (myotubes, differentiated for 2 days) and Mef2D (myotubes, differentiated for 5 days) within 2 kb around the center of the top 1000 Klf5-binding sites on day 5. The Klf5 binding sites were

*Figure 5 continued on next page*

*Figure 5 continued*

subdivided into 2 groups based on the presence of significant MyoD binding peaks in myotubes within 500 bp of the Klf5 binding regions. The distribution of Klf5 tag densities in each subgroup were shown as histogram. (C) Distribution of Klf5 tag densities in the vicinity of MyoD-bound loci in myoblasts differentiated for 0, 2 and 5 days. (D) Distribution of Klf5 tag densities in the vicinity of Mef2D-bound loci in myoblasts differentiated for 0, 2 and 5 days. (E) Venn diagram showing the overlap between the 12,769 MyoD peaks, 11,193 Mef2D peaks and 4382 Klf5 peaks identified by ChIP-seq. Note over half the Klf5 peaks (2224 out of 4382 peaks) are also bound by MyoD and/or Mef2. (F) Direct interaction of Klf5 and MyoD or Mef2 shown by co-immunoprecipitation. Whole cell extract from C2C12 cells differentiated for 5 days were immunoprecipitated using anti-MyoD or anti-Mef2 antibody. The interaction between Klf5 and MyoD or Mef2 is shown compared to the 1% Input or control samples without primary antibodies. (G) de novo motifs identified in regions bound by Klf5 in myotubes after differentiation for 5 days. (H). HEK-293T cells were co-transfected with *Myog*-luciferase; Klf5, MyoD, and Mef2D expression plasmids, as indicated; and a control CMV-renilla luciferase plasmid. Luciferase activities relative to the basal level of *Myog*-luciferase cotransfected with empty expression vectors are shown. Data represent means ± SEM. (**p<0.01, ***p<0.001, ****p<0.0001). Representative data from three individual experiments are shown.

The following figure supplements are available for figure 5:

**Figure supplement 1.** Klf5, MyoD and Mef2D are colocalized in the myogenesis-related gene loci.

**Figure supplement 2.** MyoD and Klf5 are recruited to the E-box-containing enhancer of the *Myog* locus.

**Figure supplement 3.** Klf5 binding regions in undifferentiated myoblasts.

results suggest that Klf5 has different targets and functions in myoblasts vs. myotubes. Because this study focuses on Klf5's function during muscle differentiation and regeneration, we will characterize Klf5 binding mainly in myotubes.

Analysis of Klf5-bound motifs revealed that the top motif among the Klf5 cistrome matches the consensus of Krüppel-like factor (*Figure 5G*). The Myf5 (E-box consensus bound by MRF, including MyoD) and Mef2D motifs were also identified in the de novo motif analysis, suggesting that colocalization of Klf5, MyoD and Mef2 are programmed by cis-regulatory logic. These results demonstrate a significant role for Klf5 in myogenic differentiation, acting in coordination with two other myogenic transcription factors, MyoD and Mef2.

To assess the functional significance of Klf5 in the regulation of muscle-specific target genes, we performed luciferase assays (*Figure 5H*). To eliminate effects of endogenous MyoD and Mef2D expression, HEK293T cells were co-transfected with Klf5, MyoD, Mef2D and *Myog* promoter-luciferase plasmids, which contained the proximal promoter region bound by Klf5, MyoD and Mef2 determined from each ChIP-seq (*Figure 5B*). We found that Klf5 alone activated the *Myog* reporter. Moreover, Klf5, MyoD and Mef2D activated the *Myog* reporter additively (*Figure 5H*). Together, these findings indicate that Klf5 is an active transcription factor for myogenesis, cooperating with MyoD and Mef2.

## Klf5 is necessary for the recruitment of MyoD to muscle-specific target genes

Our results indicate that the Klf5 cistrome significantly overlaps those of MyoD and Mef2 (*Figure 5A–E*), and Klf5 directly interacts with MyoD and Mef2 (*Figure 5F*). In addition, the compromised myogenic phenotype observed in *Klf5* knockout SCs and C2C12 cells (*Figures 2–4*), and the reduction in the fraction of MyoD-regulated gene expression induced by *Klf5* knockdown in differentiating C2C12 cells (*Figure 4C,D*), also suggest Klf5 affects MyoD-dependent gene regulation. To test whether Klf5 is required for recruitment of MyoD to target sites, we performed ChIP-seq of MyoD in *Klf5*-null and GFP-targeted control C2C12 myotubes to determine the effects of the *Klf5* deletion on the MyoD cistrome. Although levels of MyoD and Mef2 protein were not affected by *Klf5* deletion (*Figure 6—figure supplement 1*), ChIP-seq revealed that MyoD binding was markedly decreased in *Klf5* knockout cells compared to the GFP-targeted control cells on 3 day post-differentiation, as exemplified with *Myod1*, *Myog, Myl4* and *Mybph* (*Figure 6A and B*). This decrease in MyoD binding after *Klf5* deletion was similarly observed in another set of MyoD ChIP-seqs performed independently (*Figure 6—figure supplement 2*). The impaired recruitment of MyoD to the

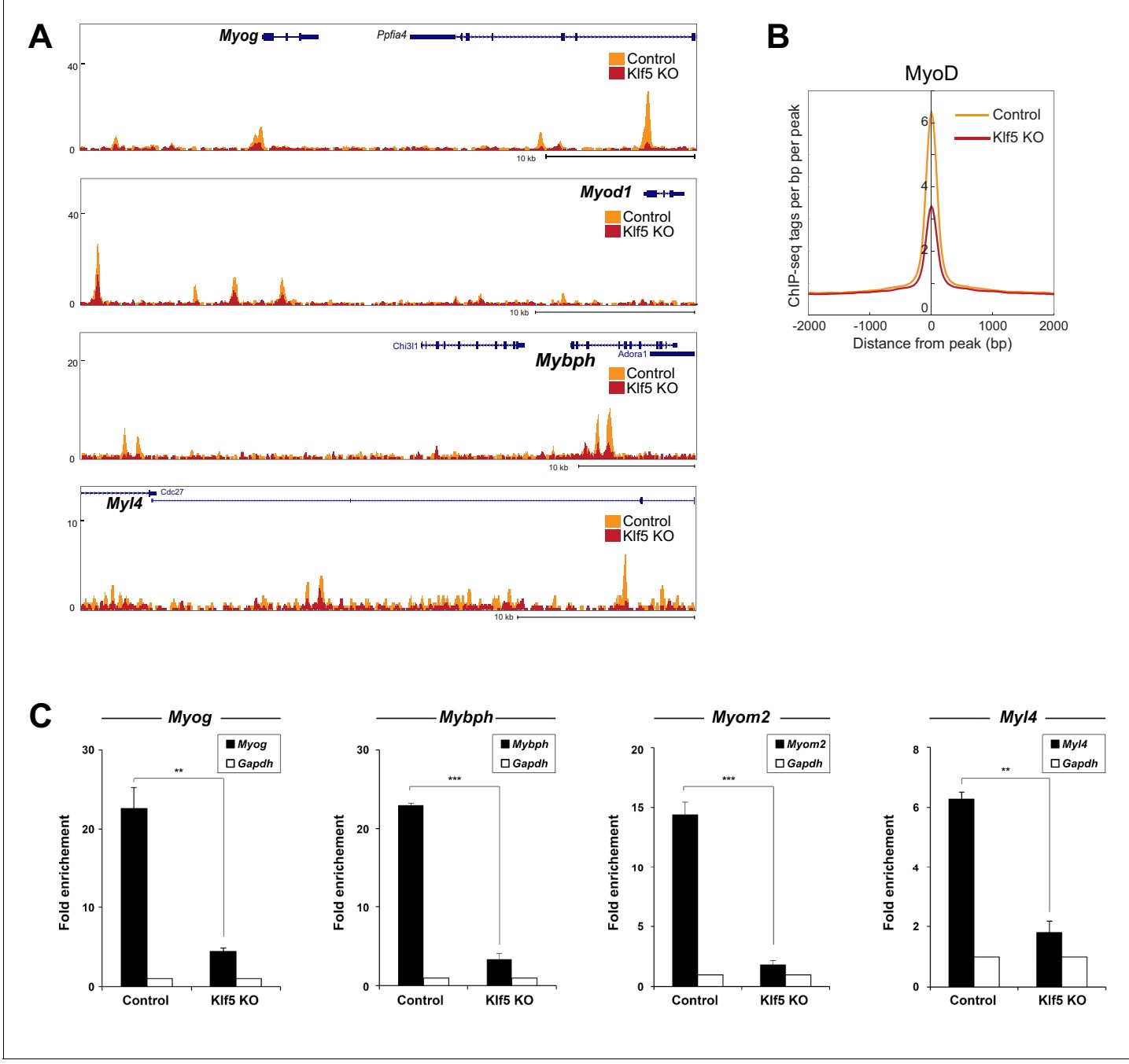

**Figure 6.** MyoD function is inhibited in *Klf5*-null C2C12 myoblasts. (**A**) UCSC genome browser images illustrating normalized tag counts for MyoD at *Myod1, Myog, Myl4* and *Mybph* gene loci in GFP-targeted control (orange) or *Klf5*-null (red) C2C12 myotubes differentiated for 3 days. (**B**) Distribution of MyoD tag densities in the vicinity of MyoD-bound enhancers in the GFP-targeted control or *Klf5*-null C2C12 myotubes differentiated for 3 days. (**C**) Comparison of MyoD recruitment at E-box containing enhancers of the *Myog, Mybph, Myom2* and *Myl4* gene loci in the GFP-targeted control and *Klf5* null C2C12 myotubes differentiated for 3 days. Data represent means ± SEM. (**p<0.01, ***p<0.001, n = 3, biological replicates).

The following figure supplements are available for figure 6:

**Figure supplement 1.** Levels of MyoD and Mef2 protein were not altered by Klf5 deletion.

**Figure supplement 2.** MyoD function is inhibited in *Klf5*-null C2C12 myoblasts -analysis in the another set of control and *Klf5*-null cells.

regulatory region of myogenic genes was further confirmed by ChIP-qPCR (*Figure 6C*). These results suggest that Klf5 plays a significant role in the recruitment of MyoD to target genes.

## Klf5 is required for satellite cell differentiation

To further test the requirement for Klf5 in SC differentiation, SCs were isolated from the EDL muscle of *Pax7^CE/+;Klf5^flox/flox* and control mice. Those SCs were cultured in growth medium for 3 days, then treated with 4OH-Tmx for 2 days to induce *Klf5* deletion by Cre-ER. In 4OH-Tmx treated *Pax7^CE/+; Klf5^flox/flox*SCs, Klf5 protein was undetectable. In these Klf5 knockout cells, myogenin protein was markedly reduced as compared to that in control *Pax7^CE/+;Klf5^flox/flox*SCs without 4OH-Tmx treatment (*Figure 7A*). In addition, mRNA expression levels of *Klf5* target genes exemplified by *Myog*, *Mybph* and *Myom2* were significantly decreased in those cells (*Figure 7B*). Collectively, these results indicate that Klf5 is indispensable for differentiation of primary SCs as well as C2C12 cells.

## Discussion

Several pieces of evidence gathered in this study clarify the essential role of Klf5 in the control of adult skeletal muscle regeneration. First, although expression of Klf5 protein remained low in Pax7[+] quiescent and activated SCs, Klf5 was significantly induced in myoblasts differentiating after skeletal muscle injury (*Figure 1*). Second, SC-specific deletion of *Klf5* markedly impaired muscle regeneration after injury due to a failure of differentiation (*Figure 2*), which confirmed the pivotal involvement of Klf5 in muscle regeneration *in vivo*. Third, suppressed differentiation into myotubes was observed in *Klf5*-null C2C12 myoblasts (*Figure 3*), after specific *Klf5* knockdown *in vitro* (*Figure 3—figure supplement 3*) and SCs from Klf5 KO mice (*Figure 7*). Fourth, inhibition of Klf5 affected transcriptional regulation of muscle-related genes by MRFs, including MyoD and myogenin (*Figures 4–6*). Collectively, these observations indicate that Klf5 is an essential component of the transcriptional regulatory network governing muscle differentiation and regeneration.

The molecular mechanism underlying muscle regeneration recapitulates many aspects of the process of muscle development. In particular, many of the transcription factors that control embryonic myogenesis also contribute to adult regenerative myogenesis. When a muscle is injured, hepatocyte growth factor (HGF), released from the basal lamina of the injured muscle triggers SC activation (*Tatsumi et al., 1998*). The activated SCs migrate and rapidly proliferate to give rise to Pax7[+]Myf5[+] cells, which activate the MRF cascade (*Cooper et al., 1999*; *Kuang et al., 2007*). Our present results indicate that Klf5 is induced in myoblast-committed, non-dividing Pax7[low]MyoD[+] SCs after they are triggered to differentiate into myocytes. Likewise, *Klf5* mRNA remained low in C2C12 myoblasts and

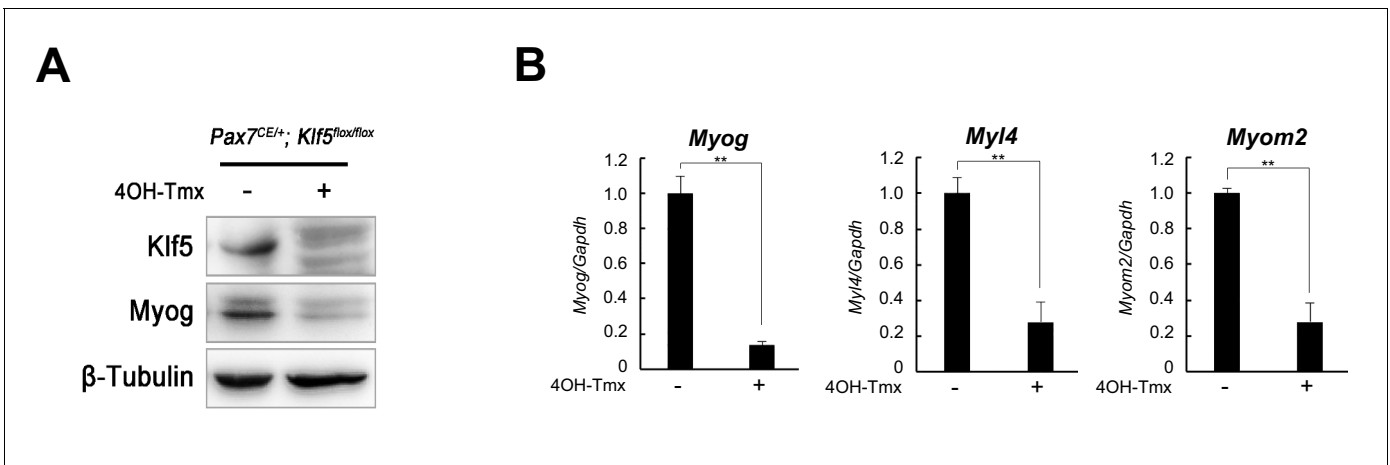

**Figure 7.** Klf5 is required for satellite cell differentiation. (**A**) Isolated SCs from Klf5 knockout mice (*Pax7^CE/+;Klf5^flox/flox*) were cultured in the presence or absence of 4OH-Tmx in growth medium for 2 days. Purified protein derived from primary myotubes differentiated for 3 days were analyzed by western blotting. Representative blots from three individual experiments are shown. (**B**) Relative mRNA expression of *Myog Myl4 and Myom2* in SCs differentiated for 3 days. Data represent means ± SEM (**p<0.01). Representative data from three individual experiments are shown.

SCs cultured in growth media. *Klf5* is greatly upregulated only after switching to differentiation media (*Figure 1C*, *Figure 3—figure supplement 1A–B*). In differentiating myoblasts Klf5 is required for expression of *Myog* and muscle-specific genes. Given that Klf5 enhances binding of MyoD to its target sites, these findings collectively suggest that Klf5 is an important regulator that links the early specification program driven by MyoD and Myf5 to the late myotube formation and maturation program driven by myogenin and Mrf4.

To further elucidate the regulatory function of Klf5 in myogenesis, it will be important to determine how Klf5 expression is induced during myocyte differentiation and regeneration. Klf5 is reportedly induced in response to external/endogenous stimuli through various signaling cascades, including p38MAPK (*Nandan et al., 2004*; *Oishi et al., 2010*), ERK1/2 (*Kawai-Kowase et al., 1999*) and IL-1β/HIF-1α (*Mori et al., 2009*). We previously showed that the p38MAPK pathway controls upregulation of Klf5 in response to angiotensin II in smooth muscle cells (*Oishi et al., 2010*). The p38MAPK pathway is activated immediately after the onset of myogenic differentiation and muscle regeneration (*Chen et al., 2007*), and is essential for myoblast differentiation (*Gonzalez et al., 2004*). Moreover, genome-wide transcriptome analysis of SCs with muscle-specific p38α deletion or C2C12 cells treated with a p38 MAPK inhibitor revealed that p38α is responsible for induction of Klf5 during myoblast differentiation (*Segales et al., 2016*). Interestingly, that study also showed that inhibiting p38 signaling does not affect expression of MyoD. It therefore seems likely that Klf5 is an essential component of the transcriptional machinery activated by p38 MAPK during myogenic differentiation. A previous study showed that Klf2 and Klf4 are induced by ERK5 signaling and contribute to muscle cell fusion (*Sunadome et al., 2011*). Microarray analyses in that study indicate that Klf5 expression is not affected by inhibition of ERK5 signaling (GSE25827). This is in agreement with our previous observations that Klf5 expression was unaffected by a MEK inhibitor in smooth muscle cells (*Oishi et al., 2010*). These findings indicate that members of Krüppel-like factor family differentially contribute to different processes during muscle differentiation and maturation. That said, it is possible that the loss of Klf5 is compensated by other Klf members, as we observed modest changes in the expression of other Klfs in Klf5-deficient C2C12 cells (*Figure 3—figure supplement 1*). Future studies of possible cooperation and competition among Klf members will be important for elucidating their functional roles in muscle biology. In addition, we found that exogenous Klf5 increased the level of eMyHC in Klf5-deleted C2C12 myotubes, but did not fully recapitulate the normal level of eMyHC expression (*Figure 3*). This insufficient rescue may reflect differences between endogenous and exogenous Klf5, including the levels and timing of Klf5 expression. In the control cells, for example, Klf5 was transiently upregulated early during differentiation, but this temporal regulation of Klf5 was lost in cells in which exogenous Klf5 was stably expressed. Future studies will also need to address the temporal regulation of Klf members during muscle differentiation and regeneration.

Our ChIP-seq results indicate that deletion of Klf5 impairs recruitment of MyoD. This demonstrates that Klf5 is required for formation of MyoD-containing transcriptional regulatory complexes in at least a subset of the MyoD cistrome (*Figure 6*). Recent studies have identified several transcription factors that interact with MyoD. For instance, Runx1 is induced in response to muscle injury and acts cooperatively with MyoD and c-Jun to regulate muscle regeneration (*Umansky et al., 2015*). TEAD4 is another factor reportedly recruited and bound to the locus also occupied by MyoD during the course of myogenic differentiation (*Benhaddou et al., 2012*). Inhibition of Runx1 or TEAD4 suppresses myogenic differentiation and regeneration, suggesting those two transcription factors, together with MyoD, are necessary to drive muscle regeneration. Of interest, we observed the sequence motif bound by Runx to rank 4th and the MCAT motif bound by TEAD4 to rank 6th among the de novo motifs identified in regions bound by Klf5 in myotubes differentiated for 5 days (*Figure 5F* and data not shown). These data suggest that Klf5 promotes myogenic differentiation by interacting with multiple transcription factors, including MyoD, Mef2, Runx1 and TEAD4. It is likely these transcription factors are differentially regulated in response to different environmental cues and/or endogenous signaling. Accordingly, the interplay between these multiple factors would enable spatiotemporal regulation and fine tuning of muscle-related gene expression. It would be important to further clarify the logic of transcriptional regulation by these multiple transcription factors. It would also be worth noting that there appears to be a complex cross-regulation of the expression of these factors. For instance, in addition to myogenin, Klf5 appears to regulate TEAD4 because there were Klf5 binding peaks in the vicinity of the genes in our ChIP-seq, and *Klf5* knockdown reduced TEAD4 levels (data not shown). In addition, based on a published microarray analysis

of effects of an individual Mef2 isoform (GSE63798) (*Estrella et al., 2015*) and our results from smooth muscle cells (*Oishi et al., 2010*), it appears Klf5 is regulated by Mef2A. This reciprocal regulation of Klf5 and other myogenic transcription factors support the notion that Klf5 is a component of an intricate transcription factor network that regulates muscle differentiation and regeneration.

In addition to MyoD and Mef2 identified in the present study, Klf5 has been shown to interact with various other transcription factors and cofactors. For instance we showed that Klf5 interacts with C/EBBβ and δ to regulate adipocyte differentiation (*Oishi et al., 2005*), interacts with PPARδ to regulate lipid metabolic genes (*Oishi et al., 2008*), and interacts with C/EBPα to control the response to renal injury (*Fujiu et al., 2011*). Klf5 also interacts with unliganded RAR/RXR to induce phenotypic modulation of vascular smooth muscle cells (*Fujiu et al., 2005*). By taking advantage of this mechanism, we showed that an RAR agonist can inhibit induction of Klf5 target genes, thereby suppressing smooth muscle cell proliferation and neointima formation. We may therefore be able to manipulate the muscle regeneration program by altering the activity of Klf5 through its interacting partners. In sum, our results provide evidence that during skeletal muscle regeneration and myoblast differentiation, Klf5 regulates the expression of late muscle-specific genes in concert with MyoD and Mef2. This finding suggests that it will be of interest to investigate this pathway further with respect to the control of muscle regeneration. It would also be interesting to know whether Klf5-mediated pathways are involved in such pathological conditions as sarcopenia.

## Materials and methods

### Mice

$Klf5^{flox/flox}$ mice (*Takeda et al., 2010*) were crossed with $Pax7^{CreERT2/+}$ mice (Abbreviated $Pax7^{CE/+}$ [*Lepper et al., 2009*]) to generate $Pax7^{CE/+};Klf5^{flox/flox}$ mice. All mice used in this study had a C57BL/6J genetic background, were between 8–12 weeks old and had age-matched littermate controls. Animal experimentation was approved by the Experimental Animal Care and Use Committee of Tokyo Medical and Dental University (approval numbers 2013-027C12 and 0170280C).

### Muscle regeneration

Tamoxifen (Sigma-Aldrich, St. Louis, MO) was dissolved in corn oil (Nakarai, Tokyo, Japan) to a concentration of 20 mg/ml. Eight- to twelve-week-old $Klf5^{flox/flox}$ and $Pax7^{CE/+};Klf5^{flox/flox}$ mice were intraperitoneally injected with 100 µl of tamoxifen solution daily for 5 days prior to induction of muscle injury. To induce muscle regeneration, 100 µl of 10 mM cardiotoxin (CTX: Sigma-Aldrich) were injected intramuscularly into the tibialis anterior (TA) muscle of anesthetized mice using a 29G syringe. Regenerating muscles were isolated 4, 7 and 28 days after CTX injection, immediately frozen in cooled isopentane in liquid nitrogen, and stored at −80°C before being cryosectioned. To analyze the distribution of fiber diameters, at least three sections from each of four $Pax7^{CE/+};Klf5^{flox/flox}$ mice and four $Pax7^{+/+};Klf5^{flox/flox}$ (control) mice were examined. Immunofluorescent images of laminin were acquired, after which the area surrounded by the laminin signal in each cross-section was quantified using Olympus cellSense Digital Imaging software. In each mouse, the diameters of 1000 (for CTX day 7) or 500 (for CTX day 28) regenerating fibers that had centrally located myonuclei were determined.

### Satellite cell isolation, culture conditions and transfection

Extensor digitorum longus (EDL) muscles were isolated from wild type C57/BL6 males and digested in type I collagenase (Worthington Biochemical Corp., Freehold, NJ). SCs were obtained from isolated myofibers by trypsinization in 0.125% trypsin-EDTA solution for 10 min at 37°C with 5% $CO_2$. SCs were cultured in GlutaMax DMEM (Life Technologies, Grand Island, NY) supplemented with 20% fetal bovine serum (FBS: Hyclone, Thermo scientific, Hudson, NH), 1% chick embryo extract (US Biological, Salem, MA), 10 ng/ml basic fibroblast growth factor (Cell Signaling Technology, Beverly, MA), and 1% penicillin-streptomycin at 37°C with 5% $CO_2$. Myogenic differentiation was induced in GlutaMax DMEM supplemented with 2% horse serum and 1% penicillin-streptomycin at 37°C with 5% $CO_2$.

To assess satellite cells obtained from *Pax7$^{CE/+}$;Klf5$^{flox/flox}$* mice, the cells were treated with 1 μM 4-hydroxytamoxifen (4OH-Tmx; Sigma-Aldrich, H7904) or vehicle 3 days after isolation, then grown in growth medium for another 2 days.

C2C12 mouse myoblasts (RRID:CVCL_0188) were purchased from the ATCC (ATCC #CRL-1772, passage number 6–8, Rockville, MD). The identity was not authenticated by our hands. Cells were free from mycoplasma contamination confirmed by monthly tests for mycoplasma. Cells were cultured in DMEM (Nakarai) containing 10% FBS and 1% penicillin-streptomycin at sub-confluent densities. C2C12 cells were differentiated into myotubes by replacing growth medium with medium containing 2% horse serum with antibiotics (differentiation medium, DM). To knock down Klf5 in C2C12 cells, siKlf5 (Thermo scientific, Dharmacon, siRNA-SMARTpool, M-062477-01-0005) treatment was carried out using RNAiMAX (Life Technologies) according to the manufacturer's instructions. Each experiments were biologically replicated at least three times.

## Generation of *Klf5* knockout C2C12 cells

*Klf5* gene was engineered as described previously (*Ran et al., 2013*). Briefly, a plasmid vector containing clustered regularly interspaced short palindromic repeats (CRISPR)-Cas9 endonuclease coupled with paired guide RNAs flanking the mouse *Klf5* gene was transiently transfected into C2C12 cells. Deletion of targeted loci was confirmed by sequencing. Cells transfected with a plasmid vector carrying CRISPR-Cas9 endonuclease coupled with paired guide RNAs flanking the GFP sequence were used as a control (Abbreviated GFP-targeted control).

## Immunofluorescent staining

Cryosections (10 μm thickness) were fixed in 4% paraformaldehyde (PFA)/PBS for 10 min at room temperature. For antigen retrieval, the sections were incubated in 0.01 M citric acid solution (pH 6.0) for 10 min at 80°C. The sections were then permeabilized in methanol for 6 min at −20°C. Cultured cells were also fixed in 4% PFA/PBS for 10 min at room temperature and permeabilized in 0.5% TritonX-100/PBS solution for 5 min at room temperature after washing 3 times. The sections and cells were then blocked in 5% BSA/PBS blocking solution, after which primary antibodies prepared in blocking solution were added and incubated at 4°C overnight. This was followed by 3 washes in PBS and incubation with Alexa fluor-conjugated antibodies (Life Technologies) in blocking solution for 1 hr at room temperature. After another 3 washes in PBS, the sections were counterstained with Hoechst 33342 solution at room temperature for 3 min. The slides were mounted with a coverslip and Fluoromount-G (SouthernBiotech, Birmingham, AL). Immunofluorescent signals from stained sections and cells were captured using a LSM710 confocal imaging system (Carl Zeiss, Inc., Oberkochen, Germany) or OLYMPUS IX73 fluorescence inverted fluorescence microscope with an Olympus DP80 camera and Olympus cellSense Digital Imaging software (Tokyo, Japan).

## Retroviral infection and plasmid vectors

Murine *Klf5* cDNA was cloned into the retroviral backbone pMX-GFP (Cell Biolabs, San Diego, CA) to produce pMX-Klf5-GFP (RV-Klf5), after which the retroviruses were packaged into the Platinum-E (PLAT-E) Retroviral Packaging Cell Line (Cell Biolabs) according to the manufacturer's instructions. Retroviral infection was accomplished at 37°C overnight using PLAT-E supernatant supplemented with 4 μg/ml polybrene.

## Immunoprecipitation

Cellular protein (600 μg) from differentiated C2C12 myotubes was mixed with 2 μg of anti-MyoD, anti-Klf5 or anti-Mef2 antibody and incubated overnight at 4°C. Twenty μl of protein G Dynabeads (1001D; Life Technologies) were then added to each sample and incubated for 1 hr at 4°C. After the incubation, the samples were washed 6 times with wash buffer (10 mM Tris-HCl, 100 mM NaCl, 1 mM EDAT, 1 mM DTT, 0.5% NP40 and 0.5% TritonX-100), resuspended in SDS sample buffer and heated at 98°C for 5 min prior to electrophoresis. Each experiments were biologically replicated at least three times.

## Immunoblotting

Proteins were quantified using a BCA Protein assay kit (Pierce, Rockford, IL) following the manufacturer's protocol. Thirty μg of total protein were separated by 10% SDS-PAGE and transferred onto PVDF Immobilon-P membranes (Millipore, Billerica, MA). Western blotting was performed using ECL Prime detection reagent (GE, Waukesha, WI) according to the manufacturer's instructions. Each experiments were biologically replicated at least three times.

## Real-time PCR analysis

Total RNA was isolated from cultured cells using an RNeasy Mini Kit (Qiagen, Valencia, CA) according to the manufacturer's protocol. RNA or ChIP-DNA were analyzed using real-time PCR with a KAPA SYBR Fast qPCR Kit (Kapa Biosystems, Woburn, MA). The primers for qPCR are listed in Table 2. Each experiments were biologically replicated at least three times.

## Luciferase assay

Murine *Klf5* (Genbank accession number: NM_009769) was sub-cloned into p3xFlag-CMV-7.1 vector. The Myog-Luciferase vector was kindly provided by Dr. T. Sato (Kyoto Prefectural University of Medicine, unpublished). The *Myod1* expression vector was a gift of Dr. P. Maire (Institute Cochin) (*Santolini et al., 2016*). The *Mef2d* expression vector was kindly provided by Dr. E. Olson (University of Texas Southwestern Medical Center). The 293T cells were transfected with a CMV-Renilla luciferase control reporter plasmid along with Myog-Luciferase, *Klf5, Myod1* and *Mef2d* expression plasmids. The total amount of plasmid DNA was kept constant using p3xFlag-CMV-7.1 empty vector. Forty-eight hours after transfection, the cells were lysed in Passive Lysis Buffer (Promega) and shaken for 15 min. The lysates were used for measurement of firefly and Renilla luciferase activities using EnSpire (PerkinElmer, Waltham, MA), after which the firefly luciferase activity was normalized to the Renilla luciferase activity.

## Library preparation and RNA-seq

PolyA-tagged RNA was pulled down from total RNA using oligo-dT magnetic beads. RNA-seq libraries were multiplexed and prepared according to the manufacturer's protocol (NEB cat#E7530, Ipswich, MA). Libraries were paired-end sequenced on a HiSeq2500 sequencer (Illumina, San Diego, CA). Reads were aligned to the mm9 genome using the default parameter for RNA-star. Aligned read files were analyzed with HOMER (http://homer.salk.edu/homer/) to calculate RPKM from the gene bodies of RefSeq genes and perform motif analysis. Reads counts were normalized using DESeq2 (*Love et al., 2014*).

## Chromatin immunoprecipitation (ChIP)

Undifferentiated C2C12 myoblasts and myotubes differentiated for 2 or 5 days were used for Klf5 ChIP. For MyoD ChIP in CRISPR-engineered cells, cells differentiated for 3 days were used. Cells were fixed in 1% formaldehyde for 10 min at room temperature. The reactions were stopped by adding glycine to a final concentration of 0.125 M and incubating for 10 min at room temperature. After washing with PBS, the cell pellet was lysed in ice-cold Cell Lysis Buffer, incubated on ice for 10–20 min and centrifuged at 3000 rpm for 5 min at 4°C. The pellet was further lysed in ice-cold RIPA buffer and incubated on ice for 10 min. The resultant cell lysate was sonicated using an ultrasonicator (output 50, 30-s pulses then 1-min pause × 13 times; UD-100: TOMY Seiko, Tokyo, Japan). The chromatin was then pre-cleared by incubating with Sepharose CL-4B (Sigma- Aldrich) for 2 hr at 4°C with constant rotation. After centrifugation, anti-MyoD or anti-Klf5 antibody was added and rotated overnight at 4°C. Protein A-Sepharose 4B fast flow (BD) was then added and incubated for 1 hr at 4°C with constant rotation. Thereafter, the beads were washed 4 times with RIPA buffer, 6 times with LiCl buffer and 3 times with TE buffer, then eluted with 2% SDS solution. After reverse cross-linking at 65°C overnight, the chromatin immunoprecipitate and Input were treated with RNase (37°C, 1 hr) and Proteinase K (42°C, 1 hr). The recovered chromatin was purified using a MiniElute PCR Purification kit (Qiagen) and used for real-time PCR analysis.

## Library preparation and ChIP-Seq

Sequencing libraries were prepared from collected DNA by using NEB next Ultra DNA library prep kit for Illumina according to the manufacturer's protocol (NEB, Ipswich, MA). Libraries were PCR-amplified for 12–15 cycles, size selected by gel extraction and sequenced on a Hi-Seq 1500 (Illumina) for 51 cycles.

## ChIP-seq analysis

ChIP-seq data for MyoD and Mef2D binding in C2C12 cells (SRA010854 and SRP017715) were downloaded from NCBI. Reads were mapped to the mm9 genome using STAR (*Dobin et al., 2013*). Peak calling and annotation was performed using HOMER version 4 (*Heinz et al., 2010*) for MyoD and Klf5, and using MACS2 (*Zhang et al., 2008*) for Mef2D. Peaks that overlapped blacklisted regions (*ENCODE Project Consortium, 2012*) or simple repeat regions were removed. Homer-identified MyoD and Klf5 peaks with scores ≥20 were considered high confidence binding sites and used for further analyses, except for detection of MyoD peaks in the vicinity of Klf5 peaks in *Figure 5*. To make a set of potential MyoD-regulated genes, RefSeq genes that contained MyoD peaks within a region −1500 bp from TSS to +1500 bp from TTS were first chosen. Then genes whose normalized counts were increased ≥2-fold in C2C12 myotubes relative to those in myoblasts were selected. The resultant MyoD target gene set contained 753 RefSeq genes. This gene set was highly enriched in GO terms related to muscle differentiation. All RNA-seq and ChIP-seq data are available in the GEO under the accession number GSE80812. To identify enriched motifs in ChIP-seq peaks, we used Homer's findMotifsGenome.pl software (*Heinz et al., 2010*).

## Statistical analysis

Sample sizes were not based on power calculations. No animals were excluded from analyses. Comparisons between two groups were analyzed using two-tailed Student's *t*-test. Differences among more than two groups were analyzed using one-way ANOVA followed by Tukey-Kramer post-hoc tests. Values of $p < 0.05$ were considered statistically significant, except in analyses involving RNA-seq and ChIP-seq. All data are means ± SEM.

## Additional information

Details of antibodies and sequences for qPCR are described in *Supplementary file 1*.

# Acknowledgement

We are grateful to Dr. A Aiba (The University of Tokyo) and Dr. Y Nakayama for providing pSpCas9 plasmid, Dr. P Maire and Dr. I Sakakibara for providing Myod1 expression plasmid, and Dr. T Sato for providing Myog-Luciferase plasmid. We thank Dr. M Kanagawa and Dr. S Fukada for the technical advice for *Pax7*$^{CreERT2/+}$ mice and Dr. S Takeda and Dr. N Ito for the technical advice for SC culture. We thank Mrs N Yamanaka and Mrs M Hayashi for their excellent technical assistance (The University of Tokyo). We thank our colleagues in the Oishi Lab and members of the Manabe Lab for discussion. This work was supported in part by JSPS KAKENHI Grant Number 26882020 (Grant-in-Aid for Research Activity Start-up to SH), 25H10 (to YO), and 15H01506 (to IM); AMED-CREST (to IM); and the grants from Nakatomi Health Science Foundation and the Uehara Memorial Foundation (to SH).

# Additional information

### Funding

| Funder | Grant reference number | Author |
| --- | --- | --- |
| Japan Society for the Promotion of Science | 26882020 | Shinichiro Hayashi |
| Nakatomi Foundation | | Shinichiro Hayashi |
| Uehara Memorial Foundation | | Shinichiro Hayashi |

| Japan Society for the Promotion of Science | 15H01506 | Ichiro Manabe |
|---|---|---|
| AMED-CREST | | Ichiro Manabe |
| Japan Society for the Promotion of Science | 25H10 | Yumiko Oishi |

The funders had no role in study design, data collection and interpretation, or the decision to submit the work for publication.

### Author contributions
SH, Conception and design, Acquisition of data, Analysis and interpretation of data, Drafting or revising the article; IM, FR, Analysis and interpretation of data, Drafting or revising the article; YS, Contributed to the in vitro studies, ChIP-qPCR and reviewed the manuscript, Acquisition of data; YO, Conception and design, Analysis and interpretation of data, Drafting or revising the article

### Author ORCIDs
Yumiko Oishi, http://orcid.org/0000-0002-4761-0952

### Ethics
Animal experimentation: Animal experimentation was approved by the Experimental Animal Care and Use Committee of Tokyo Medical and Dental University(approval numbers 2013-027C12 and 0170280C).

## Additional files

### Supplementary files
• Supplementary file 1. Table 1: list of oligonucleotides, Table 2: list of antibodies

### Major datasets
The following dataset was generated:

| Author(s) | Year | Dataset title | Dataset URL | Database, license, and accessibility information |
|---|---|---|---|---|
| Yumiko Oishi | 2016 | Klf5 regulates muscle differentiation via directly targeted muscle-specific genes in cooperation with MyoD | https://www.ncbi.nlm.nih.gov/geo/query/acc.cgi?acc=GSE80812 | Publicly available at the NCBI Gene Expression Omnibus (accession no: GSE80812) |

The following previously published datasets were used:

| Author(s) | Year | Dataset title | Dataset URL | Database, license, and accessibility information |
|---|---|---|---|---|
| Cao Y, Yao Z | 2010 | Genome-wide analysis of gene expression during differentiation in C2C12 cells | https://www.ncbi.nlm.nih.gov/sra?term=SRA010854 | Publicly available at the NCBI Sequence Read Archive (accession no: SRA010854) |
| Sebastian S, Dilworth FJ | 2013 | Alternate exon switching establishes a tissue-specific transcription factor to mediate temporal activation of gene expression during differentiation | https://www.ncbi.nlm.nih.gov/geo/query/acc.cgi?acc=GSE43223 | Publicly available at the NCBI Gene Expression Omnibus (accession no: GSE43223) |

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
