## [Decision Letter]

Thank you for submitting your article "Klf5 regulates muscle differentiation by directly targeting muscle-specific genes in cooperation with MyoD" for consideration by *eLife*. Your article has been reviewed by three peer reviewers, and the evaluation has been overseen by a Reviewing Editor and Janet Rossant as the Senior Editor. The following individuals involved in review of your submission have agreed to reveal their identity: Mukesh Jain (Reviewer #1); Atsushi Asakura (Reviewer #2).

The reviewers have discussed the reviews with one another and the Reviewing Editor has drafted this decision to help you prepare a revised submission.

In this study Hayashi and colleagues report the novel finding that KLF5, a zinc-finger transcription factor, regulates myogenesis by controlling muscle specific gene expression in cooperation with MyoD. The experimental work is well executed and the manuscript is clearly written. However a number of points need to be addressed, some of which require additional experimentation.

1) The authors should show that KLF5 is an active transcription factor for myogenesis when associated with MyoD and MEF2. For this, the authors should examine a reporter assay such as Myogenin-promoter-luciferase which is known to be regulated by MyoD and MEF2. Co-transfection experiments should be done for MyoG-Luc with Klf5, MyoD and MEF2.

2) The relationships between MyoD occupancy, KLF5 occupancy and gene expression changes are not well defined. For example, it appears that a substantial fraction of MyoD-regulated genes have higher expression with KLF5 reduction. Is MyoD binding increased at these genes, which are essentially ignored in the text? Moreover, the relationship between altered MyoD binding and changes in gene expression with KLF5 reduction are not fully delineated. Is reduced gene expression always associated with reduced MyoD binding?

The following comments (3-5) are related to this point:

3) In Figure 4, why use siRNA instead of the CRISPR lines, which are complete knockouts? The set of genes that are impacted by Klf5 deletion appears small to moderate in size, not "global" as stated in the text. Is there some defining feature of this set and how were the genes with increased expression with KLF5 reduction different than those with decreased expression? The increased expression set should be addressed here and elsewhere.

4) Given that the authors are using a published MyoD dataset in which a similar motif (Sp1) was defined, some comparison/explanation of the motif analysis would appear to be in order (Figure 5). In addition, the paper from which the published ChIP-seq data is extracted has various timepoints and it is not clear which time point is shown – certainly not a later timepoint, which is confusing, as the main point is that KLF5 goes up and influences MyoD over time.

5) What is the scale for relative read number in the ChIP-seq shown in Figure 6? How does the quality (and the identified MyoD binding sites) compare with the published dataset used earlier? How many days into differentiation is this? Why is MyoD binding globally down whereas only a subset of MyoD targets exhibit disrupted regulation in the RNA-seq data shown in Figure 4? What is the overlap between reduced MyoD binding sites in the Klf5 KO cells and the KLF5 ChIP-seq data sets? Could the two MyoD ChIPs (i.e. WT and Klf5 KO) be of different quality rather than reflective of biological differences?

Additional points:

6) Overall knockout and knockdown experiments were done with C2C12 cells. Please check that similar results are obtained using Klf5 KO primary myoblasts.

7) For completeness the authors should verify that MEF2 protein is also unaffected in the setting of Klf5 deletion.

8) The presence of other KLF family members might partially compensate for the differentiation defect of Klf5 KO cells. Please show data on the expression of other Klf gene family members during myogenic differentiation or at least discuss this further citing existing data. The authors mention KLF2/4 in the context of muscle fusion but there are others that have been linked to skeletal muscle biology and it is important to provide this background.

9) The murine pictures (Figure 1) do not really appear to support the conclusions that are stated in the text and in the legend. The change in Klf5 expression does not appear to be very dramatic, nor is the small set of images shown without quantification sufficient to support the notion that Klf5 expression specifically increases with regeneration and concomitant with myogenin expression *in vivo*. More quantitation is necessary. The C2C12 data are much more clear.

10) In relation to Figure 2, It is stated in the text that no effect of tamoxifen is seen until 28 days without injury. It would appear critical to show the effect of losing KLF5 on normal muscle morphology at this time point. There is also no methodology described for how muscle size was quantified in terms of mouse number, sections examined etc., and there are no SDs shown on the graph. Given the heterogeneity in mouse injury from experiment to experiment, "n of 3" is a limited number to conclude that "representative data" are shown. In addition, no data are shown to verify that injury was similar at early time points in a set of control and Klf5 KO mice..

11) The level of MHC expression by immunoblot (Figure 3) seems very low in the normal cells in comparison to the robust immunostaining. It is also surprising that no MHC is seen in the Klf5 KO cells, given the obvious (albeit reduced) MHC in the KO cells with immunostaining. This raises a concern about how representative the depicted blots are of MHC expression, a concern that is furthered by the clear expression of MHC in the KO cells on 3E. In addition, although the rescue of MHC expression by exogenous KLF5 in the CRISPR line is clear, it is also poor and incomplete and this should at least be discussed.

---

## [Author Response]

[…]

*1) The authors should show that KLF5 is an active transcription factor for myogenesis when associated with MyoD and MEF2. For this, the authors should examine a reporter assay such as Myogenin-promoter-luciferase which is known to be regulated by MyoD and MEF2. Co-transfection experiments should be done for MyoG-Luc with Klf5, MyoD and MEF2.*

We greatly appreciate the reviewer’s valuable suggestions. We agree there is a need for functional assays. We have performed reporter assays by transfecting a *Myogenin* promoter-luciferase construct along with Klf5, MyoD and Mef2D expression plasmids, as suggested by the reviewer. This reporter construct is composed of ~3.8 kb of the myogenin 5’-flanking region, which contains the region bound by Klf5, MyoD and Mef2D, as indicated in Figure 5. The reporter assay showed that the *Myog* promoter was activated by Klf5 alone, and addition of MyD and Mef2D further activated the promoter, suggesting activating cooperation among Klf5, MyoD and Mef2D. This result is in agreement with the co-association of the three transcription factors to the *Myog* promoter (Figure 5). The observations that *Klf5* knockdown reduced expression of many MyoD-target genes (Figure 4) and that MyoD association to the target sites was reduced by *Klf5* knockout (Figure 6) suggest Klf5 mainly acts as a positive regulator of MyoD-driven gene expression and myogenesis.

*2) The relationships between MyoD occupancy, KLF5 occupancy and gene expression changes are not well defined. For example, it appears that a substantial fraction of MyoD-regulated genes have higher expression with KLF5 reduction. Is MyoD binding increased at these genes, which are essentially ignored in the text? Moreover, the relationship between altered MyoD binding and changes in gene expression with KLF5 reduction are not fully delineated. Is reduced gene expression always associated with reduced MyoD binding?*

As pointed out, expression of a number of genes was increased in *Klf5* knockdown cells on day 3. Among the 753 MyoD-target genes depicted as red points in Figure 4, expression of 55 genes was increased by >1.5-fold. As suggested, we analyzed MyoD binding to the MyoD binding sites within +/-1.5 kb of the genes that were upregulated or downregulated by *Klf5* knockdown. As shown in Figure 8, even in regions near the upregulated MyoD-target genes, MyoD density was decreased in *Klf5* knockout cells.

The precise mechanism by which expression of the genes upregulated by *Klf5* deletion was increased despite the reduced MyoD binding is not immediately clear. It is possible that other MRFs, such as Myf5, compensate for the reduced association of MyoD, as recently reported (Conerly M., Tapscott S. et al. Dev Cell36: 375, 2016). It is also possible that MyoD binding is dispensable for the activity of the enhancers. Another possibility is that MyoD negatively regulates the activity of enhancers. More importantly, proximity of an enhancer to a gene does not necessarily indicate a functional link between them.

As suggested, we analyzed MyoD binding to sites near all genes (not limited to the MyoD target genes) whose expression levels were decreased by >1.5 fold in *Klf5* knockdown cells as compared with control cells on day 3. As shown by the red dots in the MA plot of MyoD counts at MyoD sites (Figure 8), binding was reduced at the majority of MyoD peaks near the downregulated genes, though a minor fraction of sites showed increased MyoD binding.

Because proximity of an enhancer to a gene does not necessarily indicate a functional link between them, it is not trivial to identify enhancers functionally linked to genes. Accordingly, we further analyzed the relationship between Klf5 and MyoD binding regardless the distance to genes. A new heatmap for Klf5, MyoD and Mef2D binding around the center of the top 1000 Klf5 binding peaks (Figure 5) show that at the Klf5 peaks in myotubes (day 5) that had significant nearby MyoD peaks, Klf5 binding were increased during differentiation (day 0 to 5), which is evident in histograms of Klf5 binding (Figure 5). This co-association of MyoD and Klf5 is also shown by the histogram of Klf5 binding around the center of the MyoD peaks, in which Klf5 binding was increased as differentiation progressed (Figure 5) In sharp contrast, a smaller fraction of top 1000 Klf5 peaks in myoblasts (day 0) had nearby MyoD peaks, and Klf5 binding was unchanged or decreased during differentiation (Figure 5—figure supplement 3). These results suggest there are two subsets of Klf5 binding sites: those with increasing Klf5 binding during differentiation and those with decreasing Klf5 binding. The former is more strongly related to adjacent MyoD binding, while the latter is less associated with MyoD binding. Because of cooperation between Klf5 and MyoD during differentiation, *Klf5* deletion might have a stronger impact on the MyoD binding sites associated with increasing Klf5 binding. Indeed, *Klf5* deletion markedly reduced MyoD binding at the MyoD binding sites with increased Klf5 binding on day 5 as compared with that on day 0 (>4-fold increased Klf5 counts within 400 bp around the center of MyoD binding peak), whereas *Klf5* deletion did not affect MyoD binding at the MyoD sites with decreased Klf5 binding (>4-fold decreased Klf5 counts) (see Figure 8). These results demonstrate that *Klf5* deletion has a strong impact on the sites with increasing association with Klf5 and MyoD, and support the notion that Klf5 and MyoD cooperatively proceed during the muscle differentiation program. Though these analyses support our conclusions, extensive ChIP-seq and chromosome conformation capture analyses will be needed to further clarify relationships between enhancer activity and Klf5/MyoD binding, and enhancer activity and gene transcription. Moreover, the present study focuses on the Klf5’s function in myogenesis and regeneration. Accordingly, while we have added heatmaps of Klf5, MyoD and Mef2D and briefly discussed the increasing co-association of Klf5 and MyoD in differentiation, we decided not to add the analyses shown in Figure 8.

Author response image 1.(**A**) Histograms of MyoD tag densities around MyoD binding peaks proximal to MyoD target genes that were upregulated or downregulated (>1.5-fold) by *Klf5* deletion. (**B**) MA plot of normalized tag counts at MyoD biding regions in *Klf5* knockout and control cells. MyoD binding regions proximal to genes whose expression was decreased (>1.5-fold) in *Klf5* knockout cells as compared with control cells are shown as red dots. (**C**) Histograms of MyoD tag densities around MyoD binding peaks that had >4-fold increase or decrease in counts nearby (400 bp) *Klf5* between day 0 and 5. MyoD densities in *Klf5* knockout and control mice are shown.**DOI:**
http://dx.doi.org/10.7554/eLife.17462.021

*The following comments (3-5) are related to this point:*

*3) In Figure 4, why use siRNA instead of the CRISPR lines, which are complete knockouts? The set of genes that are impacted by Klf5 deletion appears small to moderate in size, not "global" as stated in the text. Is there some defining feature of this set and how were the genes with increased expression with KLF5 reduction different than those with decreased expression? The increased expression set should be addressed here and elsewhere.*

We chose to use cells transiently transfected with siRNA instead of the CRISPR lines because other transcription factors, including Klf family members, might compensate for the defect in *Klf5* in stable knockout cells, as pointed out in comment (8). We agree that the set of genes affected by *Klf5* deletion appears small to moderate in size. Our intention was to indicate that *Klf5* deletion had a wide impact on expression of genes related to myogenesis, because 1472 genes in the hallmark myogenesis gene set out of 12744 annotated transcripts were reduced >1.5-fold in *Klf5* knockdown cells, and *Klf5* deletion severely affected myogenesis. However, we absolutely agree with the reviewer that the use of “globally” may cause unnecessary misunderstanding. Accordingly, we avoided the use of “globally” and revised the sentences accordingly.

As suggested we analyzed gene set enrichment in the genes upregulated by *Klf5* knockdown. The results indicate that no gene sets directly related to myogenesis was significantly enriched. This is presented in the revised manuscript and shown in [Supplementary-material SD1-data].

*4) Given that the authors are using a published MyoD dataset in which a similar motif (Sp1) was defined, some comparison/explanation of the motif analysis would appear to be in order (Figure 5). In addition, the paper from which the published ChIP-seq data is extracted has various timepoints and it is not clear which time point is shown – certainly not a later timepoint, which is confusing, as the main point is that KLF5 goes up and influences MyoD over time.*

We thank the reviewer for comments pointing out such important issues. The MyoD ChIP-seq dataset from a previously published paper (Cao et al. Dev Cell2010) used C2C12 cells differentiated into myotubes for 48 h. We chose it to analyze the co-association of MyoD and Klf5 in myotubes, because Klf5 was increasingly recruited to many MyoD-binding regions over the course of C2C12 differentiation (please see Figure 5). This is now more clearly presented in the figure legend.

In the present study we used Homer (http://homer.salk.edu/homer) for de novomotif discovery. This program first assembles highly similar sequences as a single motif and then assigns the known motif that is most closely matched with the de novomotif. The rank 4 motif in Figure 4 most closely matched the KLF5 motif in Homer’s motif database. As pointed out, Sp and KLF family transcription factors bind to similar GC-rich sites, though their optimal binding sequences differ (Genomics87:474-482, 2006). The earlier paper (Cao et al. Dev Cell. 2010) identified a high affinity Sp1 site with a “CCGCCC” core sequence. Our rank 4 motif is GGG(C/T)G(T/G)GGCC, which may include “CCGCCC”, though the sequence better fits the known KLF binding consensus than the Sp1 motif. To better characterize the association of the Sp1 motif with MyoD binding sites, we analyzed appearance of the known Sp1 motif. The result indicates that the Sp1 motif is present in 4.9% of the MyoD-binding regions (*P*=1e-19). We added this result to Figure 4 and indicate the use of Homer’s software in the Materials and methods. We have revised the section describing the results of the motif findings to better reflect the algorithm of Homer. We also clearly state that the identified Klf5 binding sequences may contain Sp1 binding sequences.

*5) What is the scale for relative read number in the ChIP-seq shown in Figure 6? How does the quality (and the identified MyoD binding sites) compare with the published dataset used earlier? How many days into differentiation is this? Why is MyoD binding globally down whereas only a subset of MyoD targets exhibit disrupted regulation in the RNA-seq data shown in Figure 4? What is the overlap between reduced MyoD binding sites in the Klf5 KO cells and the KLF5 ChIP-seq data sets? Could the two MyoD ChIPs (i.e. WT and Klf5 KO) be of different quality rather than reflective of biological differences?*

We apologize for the insufficient description of methods and experimental conditions. We performed MyoD ChIP-seq in *Klf5*-knockout C2C12 cells generated using the CRISPR-Cas9 method. As a control, we used C2C12 cells transfected with CRISPR-Cas9 endonuclease coupled with paired guide RNAs flanking the GFP sequence (GFP-targeted control). The cells were harvested 3 days after induction of differentiation for ChIP-seq analysis. We have expanded the experimental procedure section to address these issues (subsection “Klf5 is required for myoblast differentiation”, second paragraph).

Because ChIP-seq of MyoD in *Klf5* knockout and control cell lines in the original manuscript had less than 10 million mapped reads (5,047,433 and 8,016,529 mapped reads, respectively), we repeated the sequencing to increase the sequence depth. Now the numbers of mapped reads are comparable with the previously published data (Cao Y. et al) used to identify MyoD binding regions. Still, the number of MyoD peaks (2,225) that were identified using the CRSPR-GFP-control cells, is smaller than the number of MyoD-binding peaks (12,769) identified using the published data. However, 86% of our MyoD peaks overlapped with the MyoD peaks in the published data.

Moreover, the heatmap for MyoD binding shows a strong similarity (please see Figure III). Clearly, our ChIP-seq reads are enriched at the MyoD peaks identified by the published MyoD ChIP-seq. However, the peak intensity was generally lower in our MyoD ChIP-seq than that in the published data, which appears to be one reason why our ChIP-seq identified only a subset of MyoD peaks. A number of technical differences are likely to affect the quality of ChIP-seq. For instance, the antibody we used was one commercially available from Santa Cruz and recognized the full length of mouse MyoD. By contrast, Cao et al. used three antibodies developed in their lab that recognized amino acids 160-307, 60-219 and 3-318 of mouse MyoD (Cao Y et al. Dev Cell18: 662, Tapscott, SJ et al. Science242: 405-11, 1988). This difference in antibodies likely contributes to the relatively low peak intensity in our ChIP-seq. That said, the very high overlap (86%) between our MyoD peaks and the MyoD peaks in the published data, and the very similar read distributions in the heatmap, indicate that the two antibody sets largely recognize the same regions.

Table

Mapped readsMyoD ChIP-seq (C2C12 cells differentiated into myotube for 48 h, Cao et al. Dev Cell2010)8,797,472MyoD ChIP-seq in CRISPR-GFP-control C2C12 cells differentiated into myotube for 72 h13,628,080MyoD ChIP-seq in CRSPR-KLF5 knockout C2C12 cells differentiated into myotube for 72 h15,801,588

To further confirm the observation that MyoD recruitment was impaired in the absence of *Klf5*, we performed an independent MyoD ChIP-seq experiment using another set of cells as a biological replicate. The observation that the lack of *Klf5* reduces MyoD binding was reproduced in the independent experiment, verifying that impaired MyoD recruitment reflects biological differences, but not a difference in the quality of MyoD ChIPs in control and *Klf5* knockout cells. These results are now presented in Figure 6 and Figure 6—figure supplement 2.

Among 735 MyoD target genes, the expression levels of 246 genes were decreased by >1.5 fold in *Klf5* knockdown cells. Still, as pointed out, expression of many genes was unchanged or increased. The precise reason why MyoD binding was globally decreased while expression of many MyoD target genes was not decreased is not immediately clear. One possible explanation is that other MRFs, such as Myf5, compensate for the functional defect of MyoD, as recently published by Conerly M et al. (Dev Cell36:375, 2016). It is also possible that MyoD is dispensable for enhancer activity. Among the 753 MyoD-target genes depicted as red points in Figure 4, expression of 55 genes was increased by >1.5-fold, and 246 genes were decreased by >1.5-fold. MyoD binding to regions proximal (+/- 1.5 kb of gene) to those upregulated and downregulated genes was similarly decreased in *Klf5* knockout cells (please see Figure 9). Interestingly, however, Klf5 binding to MyoD sites near MyoD-target genes upregulated by *Klf5* deletion was much lower than to sites near downregulated genes (Figure 9). This lesser association of Klf5 with MyoD-containing enhancers is also illustrated by the heatmap for Klf5 and MyoD binding around the center of the MyoD-binding sites (Figure 9). These results suggest that the enhancers near the upregulated MyoD-target genes are likely to be indirectly affected by the loss of *Klf5*, though the precise regulatory mechanisms affecting the enhancer activity are not immediately clear. Given our current findings demonstrating the mechanistic importance of Klf5 on MyoD recruitment, we feel exploring this further is beyond the scope of the present paper, and that these issues should be addressed in our future studies. We thank the reviewers for these constructive comments.

Author response image 2.(**A**) Histograms of MyoD tag densities at MyoD binding regions proximal to MyoD target genes that were upregulated or downregulated (>1.5-fold) by *Klf5* deletion. (**B**) Klf5 binding to MyoD-binding regions proximal to MyoD-target genes whose expression was downregulated or upregulated by *Klf5* deletion. (**C**) Heatmaps for Klf5 and MyoD binding to MyoD-binding regions.**DOI:**
http://dx.doi.org/10.7554/eLife.17462.022

*Additional points:*

*6) Overall knockout and knockdown experiments were done with C2C12 cells. Please check that similar results are obtained using Klf5 KO primary myoblasts.*

We greatly appreciate the reviewer’s valuable suggestions. We agree that the data obtained from primary satellite cells strengthen our observations. We have performed experiments with satellite cells isolated from*Pax7^CreERT2/+^;Klf5^flox/flox^*(satellite cell-specific *Klf5* knockout) and the control Pax7^+/+^;Klf5^flox/flox^mice. After treatment with 4OH-Tamoxifen to delete *Klf5*, *Pax7^CreERT2/+^;Klf5^flox/flox^*cells showed defective myogenic differentiation indicated by decreased Myogenin protein expression and decreased *Myog, Myl4* and *Myom2* mRNA expression. These results are now presented in Figure 7.

*7) For completeness the authors should verify that MEF2 protein is also unaffected in the setting of Klf5 deletion.*

As suggested we analyzed Mef2 protein levels. We have added new panel in Figure 6—figure supplement 1. Mef2 protein was not affected by *Klf5* deletion.

*8) The presence of other KLF family members might partially compensate for the differentiation defect of Klf5 KO cells. Please show data on the expression of other Klf gene family members during myogenic differentiation or at least discuss this further citing existing data. The authors mention KLF2/4 in the context of muscle fusion but there are others that have been linked to skeletal muscle biology and it is important to provide this background.*

We thank the reviewer for pointing out these important points. We performed qRT-PCR to detect expression of other Klf family members in *Klf5* knockdown and control cell on day 3. Levels of Klf6, Klf13, Klf15, and Klf16 expression were modestly, but significantly increased in Klf5 knockdown cells on day 3. On the other hand, expression of Klf10 and Klf11 was decreased in Klf5 knockdown cells (Figure 3—figure supplement 1).

As pointed out, previous studies have shown that Klf members play diverse and important roles. This includes Klf15 in muscle biology. We have expanded the Introduction to better introduce Klfs in the field of muscle biology by citing a recent very excellent review by Prosdocimo DA and Jain MK et al. Trends Cardiovasc Med25, 278, 2015.

*9) The murine pictures (Figure 1) do not really appear to support the conclusions that are stated in the text and in the legend. The change in Klf5 expression does not appear to be very dramatic, nor is the small set of images shown without quantification sufficient to support the notion that Klf5 expression specifically increases with regeneration and concomitant with myogenin expression in vivo. More quantitation is necessary. The C2C12 data are much more clear.*

Although we show expression of Pax7 and Klf5 in a myoblast attached to an isolated myotube in Figure 1—figure supplement 1, the lack of images of a section of muscle in a steady state that corresponds to the images shown in Figure 1 may make comparison unclear. In line with this, representative images of uninjured muscle sections were added to the revised Figure 1. Klf5 was barely expressed in the quiescent satellite cells and in myonuclei in the intact muscle (Figure 1, upper panel). We have also added a figure showing levels of Klf5 mRNA expression over the course of muscle regeneration. Klf5 expression was markedly increased along with Myogenic factors such as Pax7, Myod1, Myogenin and Myh3 during muscle regeneration (Figure 1).

*10) In relation to Figure 2, It is stated in the text that no effect of tamoxifen is seen until 28 days without injury. It would appear critical to show the effect of losing KLF5 on normal muscle morphology at this time point. There is also no methodology described for how muscle size was quantified in terms of mouse number, sections examined etc., and there are no SDs shown on the graph. Given the heterogeneity in mouse injury from experiment to experiment, "n of 3" is a limited number to conclude that "representative data" are shown. In addition, no data are shown to verify that injury was similar at early time points in a set of control and Klf5 KO mice.*

We apologize for the insufficient description of the experimental methodology. Quantification of myofibers in immunostained tissue sections was performed as below: To analyze the distribution of fiber diameters, at least three individual sections from each of four *Pax7^CE/+^;Klf5^flox/flox^*and four or five (for day7 and 28, respectively) *Pax7_+/+_;Klf5_flox/flox_*(control) mice were examined. Immunofluorescent images of laminin were acquired, after which the area surrounded by laminin signal in each cross-section was quantified using Olympus cellSense Digital Imaging software. In each mouse, the diameters of 1000 (for CTX day 7) or 500 (for CTX day 28) regenerating fibers that had centrally located myonuclei were analyzed. This is now stated in the experimental procedure.

As pointed out, we observed certain histological differences among the mice in the same experimental group. However, the difference between wild-type and *Klf5* KO mice was clearly more pronounced than differences within the group. Muscle phenotypes after CTX-mediated injury, particularly the distribution of muscle diameters was similar among each group at each time point (day 7 and day 28). To better illustrate this, we show myofiber diameter in each mouse in Figure 10. The graphs clearly show that in all three *Klf5* KO mice, the histogram of muscle diameters was shifted toward smaller diameters. Accordingly, we are confident that the muscle phenotypes we described are consistent across experiments.

Author response image 3.**DOI:**
http://dx.doi.org/10.7554/eLife.17462.023

We agree that number of animals (3) is relatively small for the muscle regeneration experiment (on day 28). We carried out the muscle regeneration experiment in another *Klf5* KO and control mice and found that muscle regeneration was markedly attenuated in *Klf5* KO mice on day 28. This result was added to the analysis of the distribution of fiber diameters and presented in Figure 2.

We assessed the morphology of *Klf5*-deleted skeletal muscle using H&E staining on day 28 after injecting tamoxifen. This confirmed that no obvious morphological change or sign of degeneration was present on day 28 after loss of *Klf5*. This result is presented in the new Figure 2—figure supplement 1.

In addition, we assessed the injured muscle on day 4 after CTX injection to verify that the injury was similarly evoked in a set of control and *Klf5* KO mice at an early time point (revised Figure 2).

Author response image 4.**DOI:**
http://dx.doi.org/10.7554/eLife.17462.024

*11) The level of MHC expression by immunoblot (Figure 3) seems very low in the normal cells in comparison to the robust immunostaining. It is also surprising that no MHC is seen in the Klf5 KO cells, given the obvious (albeit reduced) MHC in the KO cells with immunostaining. This raises a concern about how representative the depicted blots are of MHC expression, a concern that is furthered by the clear expression of MHC in the KO cells on 3E. In addition, although the rescue of MHC expression by exogenous KLF5 in the CRISPR line is clear, it is also poor and incomplete and this should at least be discussed.*

We thank the reviewer for pointing out this important issue. We checked the specificity of the antibody for MHC and repeated the Western blot. New Western blot data with longer exposure is now provided in revised Figure 3. It indicates that MHC protein is expressed but at a lower level in *Klf5* knockout C2C12 cells than in GFP-targeted control cells, which is in line with the result of the immunostaining.

As pointed out, exogenous KLF5 expression did not fully recapitulate MHC expression in the control cells. This insufficient rescue may be due to the differences between endogenous and exogenous Klf5, including expression levels and timing of Klf5 expression. In the control cells, for instance, Klf5 is transiently upregulated early during differentiation. However, this temporal regulation of Klf5 is lost in cell expressing exogenous Klf5. Although the precise mechanisms are not immediately clear, we added these possible mechanisms to the Discussion, as suggested.